# Microphysical processes involving the vapour phase dominate in simulated low-level Arctic clouds

Theresa Kiszler[1], Davide Ori[1], and Vera Schemann[1]

[1]University of Cologne, Institute of Geophysics and Meteorology, 50939 Cologne, Germany

**Correspondence:** Theresa Kiszler (theresa.kiszler@csc.fi) and Vera Schemann (vera.schemann@uni-koeln.de)

**Abstract.** Current general circulation models struggle to capture the phase-partitioning of clouds accurately, both overestimating and underestimating the supercooled liquid substantially. This impacts the radiative properties of clouds. Therefore, it is of interest to understand which processes determine the phase-partitioning. In this study, microphysical process rates are analyzed to study what role each phase-changing process plays in low-level Arctic clouds. Several months of cloud-resolving ICON simulations using a two-moment cloud microphysics scheme, are evaluated. The microphysical process rates are extracted using a diagnostic tool introduced here, which runs only the microphysical parameterisation using previously simulated days. It was found that the processes impacting ice are more efficient during polar night than polar day. For the mixed-phase clouds (MPCs) it became clear that phase changes involving the vapour phase dominated in contrast to processes between liquid and ice. Computing the rate of the Wegener-Bergeron-Findeisen process further indicated that the MPCs frequently (42 %) seemed to be glaciating. Additionally, the dependence of each process on the temperature, vertical wind and saturation was evaluated. This showed that especially the temperature influences the occurrence and interactions of different processes. This study helps to better understand how microphysical processes act in different regimes. It additionally shows which processes play an important role in contributing to the phase-partitioning in Arctic low-level mixed-phase clouds. Therefore, these processes could potentially be better targeted for improvements in the ICON model that aim to more accurately represent the phase-partitioning of Arctic low-level mixed-phase clouds.

## 1 Introduction

Several studies (Ebell et al., 2020; Shupe and Intrieri, 2004; Curry and Ebert, 1992) showed the importance of clouds for the Arctic radiative budget. Clouds further play a role in different feedback mechanisms, for instance, the cloud-phase feedback (Mitchell et al., 1989), and either amplify (positive feedback) or dampen (negative feedback) the warming. As the Arctic is warming up to four times faster than the global average (Rantanen et al., 2022), it is of interest to which extent clouds play a role in this. Currently, the question still remains though whether the total cloud feedback is positive or negative (Middlemas et al., 2020; Goosse et al., 2018). A specific challenge is the cloud-phase feedback as the changes in the cloud's phase-partitioning impact the cloud radiative effect (Mitchell et al., 1989; Storelvmo et al., 2015). The reason for these uncertainties regarding the cloud feedbacks stems from the difficulties of models on all scales to represent clouds, especially mixed-phase clouds (MPCs) accurately (Kay et al., 2016; Zelinka et al., 2020). These difficulties are connected to the complexity of microphysical

processes in clouds and their parameterisations in models. One consequence is that many models are unable to capture the phase-partitioning in clouds correctly (Tan et al., 2016). As a result, it is hard to quantify cloud feedbacks causing uncertainties in the climate projections (Zelinka et al., 2020).

Some models struggle to represent supercooled liquid in MPCs and often underestimate it (Cesana and Chepfer, 2012; Kiszler et al., 2023a). Huang et al. (2021) link this underestimation of cloud liquid in the CESM1 model to the Wegener-Bergeron-Findeisen (WBF) process, where ice grows at the expense of liquid water due to the lower saturation required above frozen surfaces. While sometimes the limited spatial and temporal resolution can cause the full glaciation of a cloud (Storelvmo and Tan, 2015), other studies showed the importance of the ice nucleation and WBF process to estimate the climate forcing of

MPCs correctly (Shaw et al., 2022; McGraw et al., 2023). In contrast, Zhang et al. (2020) found an overestimation of liquid in the E3SM model after several changes, including a switch in the ice nucleation scheme and microphysical parameterisation scheme. Again, this shows that the phase-partitioning and representation of cloud microphysical processes remain a challenge (Korolev et al., 2017; Morrison et al., 2020).

To understand where these uncertainties come from and address them, many studies have used sensitivity tests by varying process parameters or aerosol concentrations, where the subsequent changes of the cloud macro- and microphysics and other model components are evaluated (e.g. Lasher-Trapp et al., 2018; Shaw et al., 2022). This can provide valuable insights but makes it hard to untangle the exact contribution of each process. Additionally, the number of feasible model runs cannot cover the full range of possible parameter changes and combinations. Another approach aims to evaluate the microphysical process

rates directly. This was done, for instance, by Gettelman et al. (2013) for a general circulation model (GCM) to look at the relative importance of microphysical processes in climate models using daily rates. In a recent paper, Barrett and Hoose (2023) used so-called microphysical pathways, which include different sets of microphysical processes, to study an idealized deep convective system. Kalesse et al. (2016) found a strong connection between the deposition rate of snow and the snow mass mixing ratio in a case study of an Arctic low-level cloud. Fan et al. (2017) studied the effect of changing ice nucleating particles

(INP) and cloud condensation nuclei (CCN) concentrations on process rates, finding an increase in condensation, evaporation, deposition, sublimation and riming with increasing aerosols. All such studies are limited by different factors, such as time and spatial resolution, idealization, or a low number of cases. So far, we are lacking detailed insights into cloud microphysical process rates from atmospheric models which cover a large set of cases.

This study is motivated by the described uncertainties linked to cloud microphysics and by the lack of microphysical process rates data. To address these points we introduce a diagnostic tool which can compute the microphysical process rates of the commonly used two-moment microphysics scheme by Seifert and Beheng (2006) (Section 2.3). As model, ICON-LEM (ICOsahedral Nonhydrostatic model in the large-eddy version, Dipankar et al., 2015) is chosen, as this is a widely used model which enables cloud-resolving simulations. To reduce complexity the focus is set on Arctic low-level clouds (LLCs) at Ny-

Ålesund, Svalbard. There LLCs frequently occur (Gierens et al., 2020; Nomokonova et al., 2019). Svalbard has an above

average occurrence of MPCs (45-60 %) in comparison to the rest of the Arctic (30-50 %, Mioche et al., 2015) making the location ideal for studying the phase-partitioning of clouds.

Several months of simulations are evaluated to provide a knowledge base that is representative for Ny-Ålesund. In the results three different research questions are addressed:

1) Which microphysical processes determine the phase-partitioning in simulated MPCs and how frequently do they occur? Section 3.1

2) What role does the WBF process play in the lack of supercooled liquid water? Section 3.2

3) How do environmental conditions impact each microphysical process? Section 3.3

These questions are aimed at understanding more about known shortcomings in the cloud representation using ICON-LEM (Kiszler et al., 2023a). Additionally, the results provide detailed insights into the way each microphysical process behaves in different temperature, vertical wind and saturation regimes.

## 2 Methods and data

### 2.1 ICON simulations

The ICON-LEM (Dipankar et al., 2015; Heinze et al., 2017) simulations which we performed cover a circular domain with approximately 100 km diameter centred in Ny-Ålesund (Svalbard, 78.9 °N, 11.9 °E) and run with approximately 600 m resolution. The general setup follows the papers by Kiszler et al. (2023a) and Schemann and Ebell (2020) and a thorough evaluation of the model performance is provided in those studies. While the general performance of the model was found to be very good, there were some short comings. In Kiszler et al. (2023a) it is shown that the cloud occurrence matches the observations well but that the occurrence of liquid containing clouds is underestimated by around 30 %.

Each simulation covers 24 h, although the first 3 hours are excluded in the analysis to avoid the spin-up. The initial and boundary conditions for each ICON-LEM limited area simulation are provided by an ICON-NWP simulation with 2.4 km resolution. This ICON-NWP simulation covers a larger domain and is forced by the operational German weather service global ICON-NWP runs. The turbulence is parameterised by a 3D-Smagorinsky scheme (Dipankar et al., 2015). The two-moment scheme from Seifert and Beheng (2006) with an added hail class Blahak (2008) is used for the microphysics (referred to as SB). We use the Segal and Khain (2006) cloud condensation nuclei (CCN) activation with maritime aerosols, as well as the heterogeneous ice nucleation from Phillips et al. (2008) with the maritime aerosol concentrations. The output is given in the form of the vertical column above the grid cell containing Ny-Ålesund (meteogram) for every 9 s on 150 levels. This output includes the following hydrometeor mass mixing ratios and number concentrations: cloud droplets, rain, ice, snow, graupel and

hail.


## 2.2  Selected data

For the analysis, a subset of the data, which only includes low-level clouds, was created. This was done by first selecting all grid boxes which are cloudy using a threshold for the hydrometeor concentration of $10^{-8}$ kg kg$^{-1}$ (same as in Kiszler et al., 2023a; Schemann and Ebell, 2020) above which a grid box is defined as cloudy. For a cloud to be classified as low-level, the cloud top height must be below 2.5 km (same as Gierens et al., 2020; Chellini et al., 2022). Additionally, precipitating hydrometeors are not differentiated from non-precipitating ones. Therefore, a cloud with rain, graupel, hail or snow that reaches the ground will have a cloud base height at the ground. Further, if there is a cloud with a cloud top height higher than 2.5 km above the low cloud, we only use these cases if the higher cloud's bottom height is at least 500 m higher than the low cloud's top height.

The frozen and liquid hydrometeors are grouped in the analysis to focus on phase transitions. The frozen mass mixing ratios ("frozen mass", kg kg$^{-1}$) is the sum of cloud ice, graupel, hail, and snow, and the liquid mass mixing ratios ("liquid mass", kg kg$^{-1}$) is the sum of cloud droplets and rain. Generally, the liquid and frozen mass mixing ratios lie between $10^{-8}$ and $10^{-3}$ kg kg$^{-1}$. The occurrence of low-level clouds and their composition varies between seasons (Mioche et al., 2015). Therefore, two sets of data are used. One covers the polar night (PN, November 2021 - February 2022) and one the polar day (PD, May-August 2021). In total, for the PN, there are around 26.3 days' worth of low-level clouds, and for the PD, there are around 37.9 days. For the selection of the MPCs only cloudy pixels where both the liquid and frozen mass are above $10^{-8}$ kg kg$^{-1}$ are chosen. Of the total 23.8 days' worth of MPCs, 14.1 occur during PN and 9.7 during PD.

In the analysis the influence of the temperature, vertical velocity, and ice/water saturation on different microphysical pro-
cesses is discussed. These variables were chosen as the microphysical processes are directly connected to them. As PN and PD differ in parts strongly, a short overview of the thermodynamic conditions for the selected cloudy grid points is provided here. Fig. 1 a-c show the distributions of the temperature, vertical velocity, and ice saturation for the PN and PD. The PN temperature ranges from -32 to 2 °C with the mean at -14 °C. In contrast, the much warmer PD varies less (-22 to 8 °C) and has a mean value of -2 °C. The vertical velocity is narrowly arranged around 0 m s$^{-1}$ for both PN and PD, and both the PD and PN show
low variation (standard deviation: PD 0.35 m s$^{-1}$, PN 0.29 m s$^{-1}$). Extremes, which happen very rarely, are found more in the upwards motion, with the overall maximum at 6.43 m s$^{-1}$. The saturation with respect to ice does not reach as high values during the PD as during the PN.

Of the total 26.3 days of low-level clouds during PN, almost all contain periods with frozen hydrometeors (25.4 days), and
slightly more than half contain liquid hydrometeors (15.5 days, 58 %). For the 37.9 days of low-level clouds during PD, almost all times contained liquid (96 %) while only 31 % contained ice. This is connected to the fact that liquid occurs at higher temperatures, which are more prevalent during the polar day (Fig. 1 a). Another aspect to note is that similar to Shupe et al.

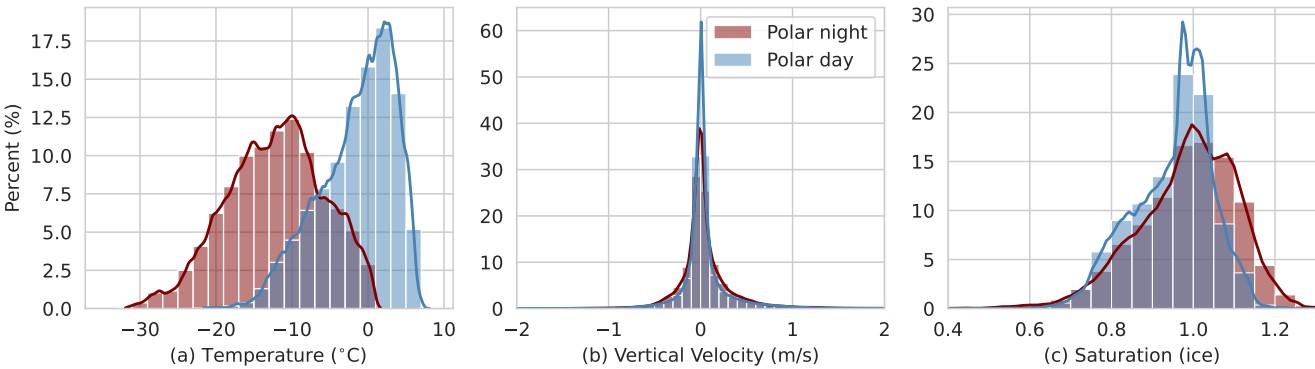

**Figure 1.** Distribution of the temperature (a), vertical velocity (b) and ice saturation ratio (c) for the polar night (PN, red) and polar day (PD, blue).

(2008) and as theorized by Korolev (2008), we found that higher upward vertical velocities are connected to higher saturation and with that also to higher hydrometeor masses (not shown). This already hints towards potential correlations between certain processes and the vertical wind and saturation which are presented in the results.

### 2.3 Microphysics parameterisation wrapper

To extract the process rates, this study uses a "microphysical parameterisation wrapper". The goal is to provide a simple diagnostic tool with spatial and temporal flexibility. Therefore, we chose to run the two-moment SB scheme and the saturation adjustment (for condensation/evaporation) independently from the model. A meteogram (single column) or 3D output file from a previous simulation can be used as input for the wrapper. The model output is provided as input to the microphysical scheme in each timestep and on the same vertical grid as used in the model. It computes a single timestep, writes out the rates and continues with the following timestep. This approach allows the use of previous simulations from which the meteogram or 3D output of the required variables exists. A flowchart is shown in Fig. A1.

This approach has the clear advantage of being very fast compared to rerunning a full simulation, and one can focus on single processes. Further, outputting an entire domain of microphysical process rates is extremely space-consuming in most cases and can be avoided by just using a spatial and/or temporal selection. Additionally, it is possible to explore potential sensitivities of microphysical processes by applying changes inside of the wrapper and using it as a test suite. As this tool is simplified and only captures a part of the model, the advantages come with some limitations. One must keep in mind that any transportation (advection and precipitation) of hydrometeors cannot be included as the model itself is not run. In this study, we are only interested in the microphysical processes. Therefore, this is not an issue.

The mass change due to a process is computed by taking the difference between the mass before and after the process and
is called ($\Delta Q_{proc}$). Here, a timestep ($\Delta t$) of 3 seconds is used for the time integration. Therefore, the process rate ($\frac{\Delta Q_{proc}}{\Delta t}$)
is given as the hydrometeor mass mixing ratio change over 3 seconds and is denoted as the "tendency" (kg kg$^{-1}$ 3s$^{-1}$) in the
following sections. A minimum threshold for the tendency of $\frac{\Delta Q_{proc}}{\Delta t} > 10^{-18}$ kg kg$^{-1}$ 3s$^{-1}$ is set to avoid including numer-
ical noise. This threshold is much lower than the threshold for the hydrometeor mass ($10^{-8}$ kg kg$^{-1}$) as the microphysical
process tendencies changing the mass can be very small. As mentioned before, the hydrometeors are summed up into a liquid
and frozen mass. The same is done with the processes, meaning, for example, that deposition is the sum of deposition onto
ice particles, snow, graupel and hail. In this study, only processes that cause phase changes are included, as we are interested
in processes that contribute to the phase-partitioning. E.g. frozen collisions are not evaluated as all frozen hydrometeors are
summed up to a single class and compensate each other. Many of the processes evaluated in this study have self-explanatory
names. Nevertheless, a brief process summary is given here and in Table 1 to prevent misunderstandings.

*Deposition and Sublimation* includes either the decrease or increase of water vapour due to phase changes between frozen
water and water vapour. They are computed by the same process and split into negative (sublimation) and positive (deposition)
contributions to the frozen mass.

*Homogeneous and heterogeneous ice nucleation* describes as one part the homogeneous nucleation of liquid aerosols, although
it is rarely cold enough (T<30 °C) in the clouds used in this study to happen. The heterogeneous nucleation describes nucle-
ation via immersion freezing and deposition nucleation. The parameterisation follows as mentioned Phillips et al. (2008). As
homogeneous cloud droplet freezing did not occur for the low-level clouds, it is neglected here.

*CCN activation* describes the activation of cloud condensation nuclei following the parameterisation of Segal and Khain (2006).

*Frozen evaporation and Melting* refer to the melting of frozen hydrometeors, which can entail evaporation, but both are treated
separately, leading to a process called "evaporation" also for the frozen hydrometeors.

*Riming* describes the accumulation of liquid mass on a frozen hydrometeor by decreasing the liquid mass and increasing the
frozen mass. In SB in ICON, this also includes the Hallet-Mossop secondary ice production. If T>0 °C enhanced melting
after riming will take place, making the frozen mass increase due to riming less as not all liquid will freeze onto the frozen
hydrometeors.

*Condensation and Evaporation* includes either the decrease or increase of water vapour due to phase changes from cloud
droplets and raindrops to vapour. They are the positive (condensation) and negative (evaporation) contributions of the sat-
uration adjustment to the liquid mass. The saturation adjustment is run once before and once after the other microphysical
processes (see Fig. A1).

*Rain freezing* only includes the freezing of raindrops and not cloud droplets. As both are summed up, though, this causes a
decrease of the total liquid mass while the total frozen mass increases.

An additional process which is not directly implemented in SB, but is analysed in this study, is the WBF process. As evap-
oration and deposition are needed simultaneously for the WBF process, it is possible to use their rates to compute the WBF
rate. The saturation adjustment, which provides the evaporation rate, is computed twice in each timestep in contrast to all other

| Process | $\Delta Q_L$ | $\Delta Q_F$ | $\Delta Q_v$ |
|---|---|---|---|
| Ice nucleation | - | + | - |
| CCN activation | + | / | - |
| Deposition | / | + | - |
| Sublimation | / | - | + |
| Evaporation | - | / | + |
| Condensation | + | / | - |
| Riming | - | + | / |
| Rain freezing | - | + | / |
| Melting | + | - | / |

**Table 1.** Impact of each process on the hydrometeor masses for liquid ($\Delta Q_L$) and frozen ($\Delta Q_F$) as well as the water vapour ($\Delta Q_v$). The liquid class contains cloud droplets and raindrops, the frozen class contains ice particles, snow, graupel, and hail. A plus indicates an increase in the hydrometeor mass, and a minus a decrease.

microphysical processes. During WBF events, the second call to the saturation adjustment happens in an atmosphere that has been deprived from moisture due to deposition on ice and hence causes additional evaporation.

## 3 Results

### 3.1 Dominating processes in low-level clouds

We used a simple but straightforward approach to understand which processes dominate the phase-partitioning in low-level clouds. For each process, the mean value over all cases was computed. The mean values can vary, for instance, with temperature, as shown in the next section, so the percentage of occurrence is used as a second metric. The outcome of this is shown for all processes in Fig. 2 split into polar night (a and c) and polar day (b and d), as well as into MPCs (a and b) and pure frozen or liquid clouds (c and d). The further a process is towards the upper right corner, the more relevant it is considered to be. As mentioned earlier, only processes which contribute to phase changes are included here. Further, we used the minimum of deposition and evaporation to compute the WBF tendency for the MPC cases.

What becomes very clear from Fig. 2 is that there is a hierarchy in how relevant a process is. In all cases evaporation seems to be strongest followed by deposition. Here, a striking difference between MPCs and single phase clouds becomes visible. While in liquid clouds it seems like the majority of the clouds are in the decaying phase, shown by the frequent evaporation (above 79 %) in contrast to condensation (below 21 %), this is not necessarily true for the MPCs. As Fig. 2 a) and b) show, deposition is stronger in MPCs in contrast to the pure ice clouds, indicating that the MPCs are generally transitioning from liquid to ice. The transition from liquid to ice via the vapour phase can be quantified using the WBF tendency which shows a frequency of around 42 % and varies little between PN and PD. At the same time, the pure ice clouds seem to be in a more

stable state although the higher frequency of sublimation indicates a slight decay also for the ice clouds. The finding that all cloud types seem to be in the process of decay, where processes acting as sinks are dominating, is potentially a local feature as only the single column of Ny-Ålesund is used here. This feature indicates that the microphysical processes may also have a strong location dependency. For instance, in Ny-Ålesund, the air over the fjord is more moist than over the land (Kiszler et al., 2023a), which may cause more evaporation over land if a cloud is advected there.

Another aspect, is that the microphysical process show different behaviours during the PN and PD. Processes involving the ice phase show a decreased mean tendency during PD in contrast to PN. Additionally, one can see that riming seems to be more frequent during the PN (30 %) than PD (21 %), while rain freezing is less frequent during the PN (PN: 32.0 % and PD: 38.64 %). Such differences between PN and PD are likely connected to the dependency of the processes on the temperature regime as discussed later. Ice and liquid formation via nucleation/activation tend to occur quite seldom, as one can see from the CCN activation (CCN act., in Fig. 2) and the homogeneous & heterogeneous nucleation (Hom. het. nuc. in Fig. 2). Using the mass change as a metric for nucleation can be misleading as the number of hydrometeors produced can say more about the impact of the nucleation process than the mass change.

Evaluating this single column, shows that microphysical processes vary strongly in their importance and depend on the location studied. It is evident that the microphysical sinks found for liquid clouds are much weaker for mixed-phase and ice clouds. Especially, for the MPCs it became clear that the WBF process acts strongly upon the liquid mass and it is therefore worth further investigating its behaviour.

## 3.2 WBF in mixed-phase clouds

The Wegener-Bergeron-Findeisen process can be a reason why models have too little supercooled liquid, impacting the representation of MPCs. As shown in Kiszler et al. (2023a) also in ICON-LEM, the amount of liquid-containing clouds is underestimated. In the previous section we have shown that the WBF process occurs very frequently in MPCs and could be a reason for the glaciation of these clouds. Therefore, this section aims to quantify and further explore the WBF process. The first investigated aspect is whether the evaporation rate increases due to the WBF process. This does not necessarily have to be the case as evaporation could just occur more frequently but does not need to be stronger. To evaluate this aspect the subselection of MPCs was evaluated where evaporation was occurring (75 % of MPC cases). This set was split into two sets. The WBF set consists of cases where deposition occurs simultaneously and where it is, therefore, sub-saturated with respect to water and saturated with respect to ice. This makes up 42 % of the MPC cases. The other set consists of cases where no deposition occurs (33 % of MPC cases). In Fig. 3 a), the distribution of the evaporation tendency for both evaporation sets is shown, and one can clearly see that they differ strongly. For the WBF cases, the evaporation tendency is generally much larger than when no WBF is occurring. It should be kept in mind that this is a logarithmic scale, where two orders of magnitude make a large difference in the amount of liquid evaporating.

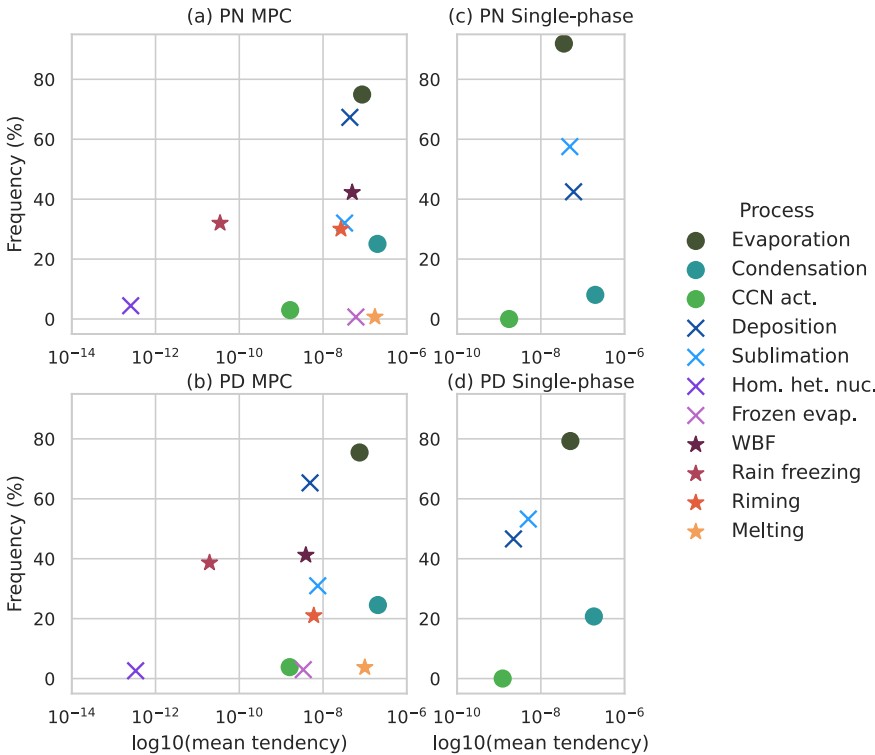

**Figure 2.** Microphysical processes mean tendencies and occurrence during the polar night (PN, a and c) and polar day (PD, b and d) shown for mixed-phase clouds (MPC, a and b) and for single-phase clouds (c and d). Dots indicate processes occurring between liquid water and vapour, crosses processes occurring between ice and vapour and stars processes between liquid water and ice.

Continuing with the impact on the total frozen and liquid mass, there too the question is whether the impact of the WBF process is significant or not. In Fig. 3 b) and c), one can see the distributions of frozen and liquid masses when deposition and no evaporation is occurring (no WBF, 25 % of MPC cases) and when deposition and evaporation are occurring simultaneously (WBF, 42 % of MPC cases). Here, too, there are clear differences between the distributions, showing a shift towards higher

frozen masses when WBF occurs. At the same time, the liquid mass distribution shifts towards lower values when WBF occurs. In Fig. 3 c), the tail of the liquid mass, which is visible for the cases where WBF occurs, is due to rain, which often occurs in subsaturated layers as it falls. Combined, this demonstrates the decrease in liquid mass while there is an increase in frozen mass when the WBF process occurs. For both Fig. 3 b) and c), we found the difference in the distributions to be statistically significant (Kruskal-Wallis test). This is also visible in the difference in mean values for both processes. When both processes

occur at the same time, the average deposition rate experiences a fourfold increase ($9.8 \cdot 10^{-9}$ to $3.9 \cdot 10^{-8}$ kg kg$^{-1}$), while the average evaporation rate also increases by around one-order-of-magnitude ($2 \cdot 10^{-8}$ to $1.3 \cdot 10^{-7}$ kg kg$^{-1}$).This shows that a significant amount of water transitions from liquid to vapour and then to the frozen phase via the WBF process.

The difference in rate change could be connected to the microphysics implementation, where the saturation adjustment is
called twice in contrast to the deposition, which is called only once, and the physics of the WBF process. Considering the typical thermodynamic situation characterizing WBF, the atmosphere is subsaturated with respect to water and supersaturated with respect to ice. This causes evaporation to occur during the first call of the saturation adjustment, providing more moisture to be deposited into ice as a result of the microphysics scheme. Then, during the second call of the saturation adjustment the atmosphere tends to return to the state it was before deposition happened. Because of this, intuitively evaporation would be
higher than deposition. Additionally, if deposition and evaporation tendencies would be the same there would be a net release of latent heat causing the equilibrium to shift towards additional evaporation. Unfortunately, it is not possible at this stage to quantify this effect in the ICON model. However, the developed microphysical wrapper can be further used to evaluate the relative contribution of the various components of the evaporation and deposition process, and assess the sensitivity of various process rates to the thermodynamic and microphysical conditions of the clouds in the model.


Several other interesting findings appeared when we looked into the question where and under what circumstances the WBF process occurs. There we looked into the temperature distribution of the WBF process and found that the WBF process seems to correlate more strongly with deposition than with evaporation (Fig. 4). Additionally, one can see that the distributions look different between the PN and PD (Fig. 4 a and b respectively). While during the PN the WBF process most frequently occurs
between -6 and -13 °C, during the PD two maxima are visible one around -3 °C and one around -10 °C. As deposition should decrease with increasing temperature (shown later), the peak at higher temperatures was not expected. The higher cloud occurrence between -5 and 0 °C could suggest that this cloud occurrence maximum causes the higher process occurrences in this range. When normalizing with respect to the cloud occurrence, we still found a slight increase of deposition and WBF around -3 °C (not shown). Therefore, we investigated whether other processes could be influencing the WBF and deposition
frequency. Indeed, it seems like riming and rain freezing play a role in the deposition and WBF rate increase during the PD at higher temperatures (Fig. 4 b). We hypothesize that the increase is due to the fact, that both riming and rain freezing increase the ice mass creating more frozen mass on which vapour may deposit. This would explain the increase of the deposition rate above -5 °C causing the WBF process to set in.

Another aspect which we evaluated was whether the deposition rate or the evaporation rate is the limiting factor for the WBF rate, as the WBF rate is based on the minimum rate of both. To explore this we used the supercooled liquid fraction (SLF, Komurcu et al., 2014) to categorize the clouds. An SLF of 1 indicates a pure liquid cloud and 0 a pure ice cloud. A mixed-phase cloud would lie between 0 and 1. Using the SLF we found that there seem to be two WBF regimes, one for clouds with high liquid amount and one for clouds with low liquid content. In cases where the liquid mass dominates, deposition is the limiting
factor for the WBF process while for low liquid mass, evaporation limits the WBF rate. This is understandable if one considers that if there is less ice available the deposition rate will be lower and if there is less liquid available there will be less mass available to evaporate (see Fig. B1 for a visualization).

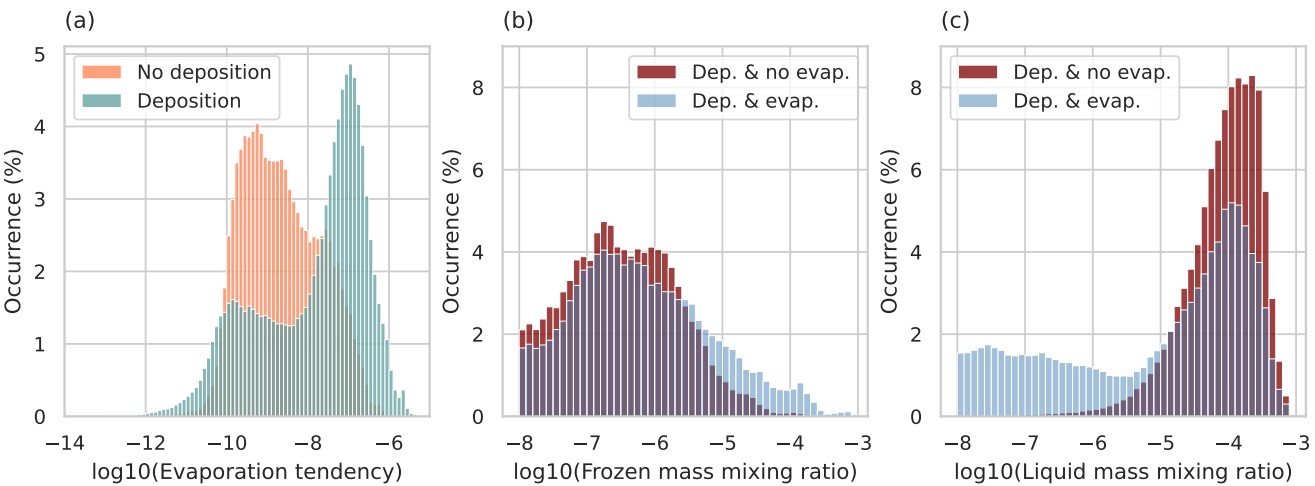

**Figure 3.** Histograms of the logarithm of evaporation tendency (a), frozen mass (b) and liquid mass (c). a) Liquid mass change due to evaporation in each timestep (evaporation tendency) for MPCs where no deposition occurs (orange, 33 %) and where deposition occurs (green, 42 %). b) Frozen mass for MPCs where deposition but no evaporation occurs (red, 25 %) and where deposition and evaporation occur simultaneously (blue, 42 %). c) is analogue to b) but shows the liquid mass.

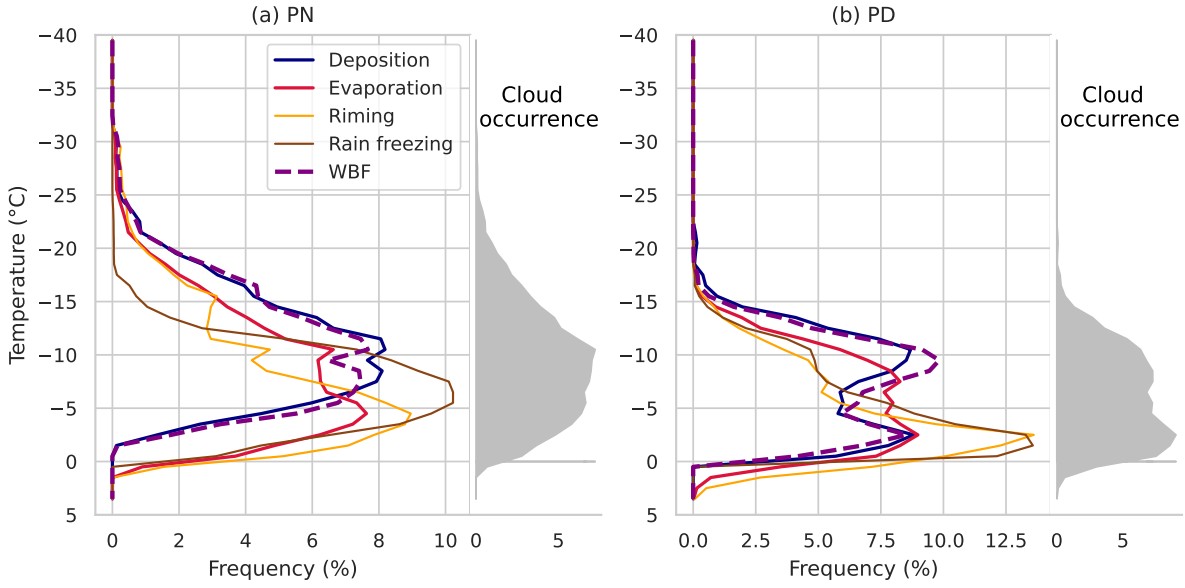

**Figure 4.** Occurrence frequency of microphysical processes with temperature shown for the polar night (PN, a) and the polar day (PD, b). Microphysical processes shown are deposition (blue), evaporation (red), WBF (purple, dashed), riming (yellow) and rain freezing (brown). The distribution of the mixed-phase clouds are shown in grey for both PN and PD. The process distributions are normalized with respect to temperature but not cloud occurrence.

### 3.3 Dependence on environmental conditions

Looking at the WBF process, and the difference of processes between PN and PD indicates that thermodynamic conditions influence the microphysical processes. To understand more about these dependencies, the microphysical processes behaviours with temperature, vertical velocity and saturation (ice, liquid water) were evaluated. It is worth mentioning here that sublimation and deposition can not occur at the same time as they are calculated by the same process. The same is true for evaporation and condensation. Other processes can occur at the same time.

Starting with the temperature dependency, it can be seen that deposition occurs relatively consistently at all temperatures below 0 °C (Fig. 5 f) while the mean mass change decreases (Fig. 5 a). Sublimation shows a similar behaviour (appendix Fig. C1 a and f), although deposition shows a slight maximum between around -10 and -20 °C where sublimation has a minimum. Another process showing a decreasing tendency with temperature is rain freezing, which occurs more often for higher temperatures, but the amount of frozen mass decreases with temperature (5 e and j). Rain freezing, as expected, is more efficient at lower temperatures but has less total impact the colder it is. Interestingly, a bi-modal distribution is visible in Fig. 5 d and i for riming for both the tendency as well as the occurrence. One maximum lies below approximately -20 °C, where there are altogether few cases, and one above -10 °C. This is possibly connected to the maximum saturation difference between ice and liquid water as the cloud droplet mean mass has a local minimum of around -18 °C. This hypothesis would be supported by the maximum of deposition in regions where riming is lowest.

Of the processes affecting the liquid mass, evaporation dominates throughout all temperature ranges where liquid occurs. Both the occurrence and the tendency of evaporation increase with increasing temperatures (Fig. 5 b and g). Of all processes, evaporation shows the largest tendency spread. This potentially indicates that evaporation may be more strongly influenced by other factors at negative temperatures, in contrast to other processes which depend more clearly on the temperature. Combining evaporation and deposition, the WBF process occurrence has a clear maximum between -10 and -5 °C (Fig. 5 h). At the same time WBF seems to have the highest tendency for values below -20 °C, although here again caution is required due to the number of cases (Fig. 5 c).

Further processes, shown in the appendix, include condensation, which has a relatively constant tendency and increases with temperature in occurrence, and CCN activation which occurs rarely and its occurrence with temperature is similar to condensation (Fig. C1). Melting and evaporation, due to melting, are only active above 0 °C and decrease accordingly with increasing temperatures as less and less frozen mass is available (Fig. C1). Homogeneous and heterogeneous ice nucleation occurs very rarely and more cases would be required to properly describe its thermodynamic dependencies (Fig. C1).

The next variable to look at is the vertical velocity. This section only focuses on processes where a signal can be seen. One process is riming, which increases with upward velocity in occurrence. This can mainly be seen during the PN as riming is

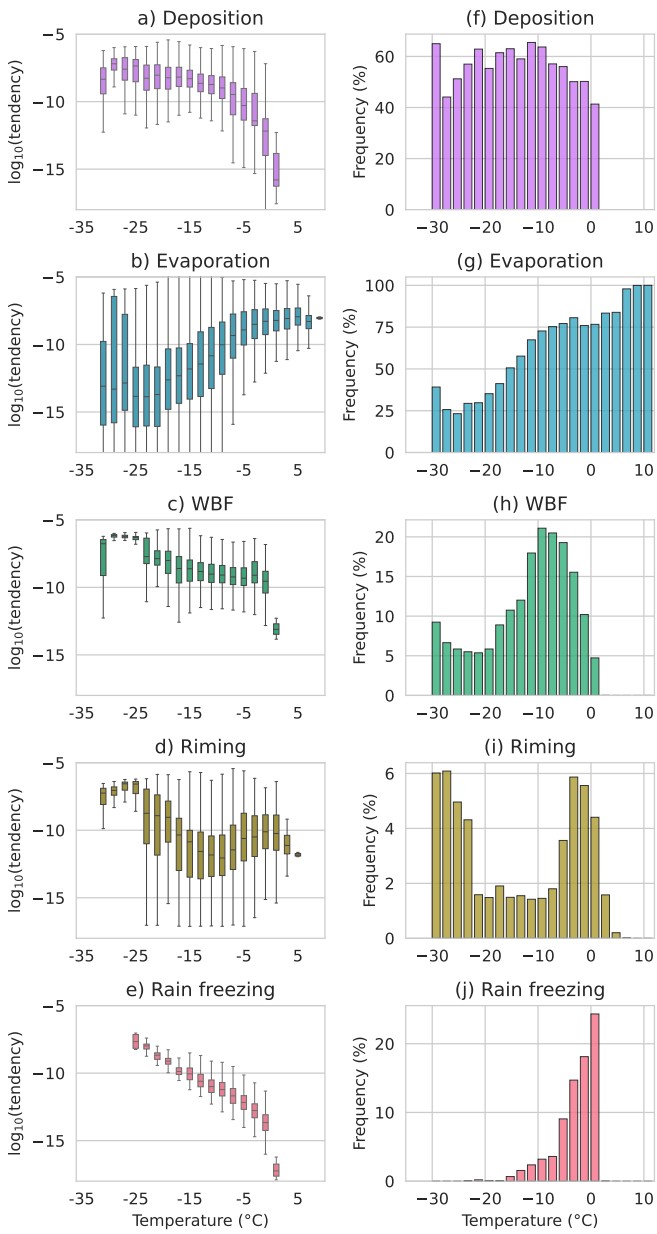

**Figure 5.** Temperature dependence of microphysical processes. Left column: box plots for temperature bins. Right column: occurrence for each temperature bin. The data includes the polar night and polar day, and bins of 2 °C are used.

much more frequent there (Fig. 6 b). The riming tendency may suggest an increase with upwards velocity as shown in Fig. 6 a), although the fact that only 1 % of the cases are above $1.2 \text{ m s}^{-1}$ makes this slightly speculative. If one discards the lowest and highest 1 % of the vertical wind speed, then a decrease of sublimation with upward velocity and its increase towards higher downward velocity can be seen (Fig. 7 a, white areas). For sublimation, no difference in the behaviour between polar night and day is found. For deposition, one would expect the opposite behaviour, but as visible in 7 b), combining the PD and PN, such behaviour is not completely obvious.

Interestingly, for deposition, a difference in its behaviour with vertical velocity can be seen between PN and PD. The deposition frequency increases with upward velocity clearly for the PN (Fig. 7 d). This behaviour is not so clear for the PD (Fig. 7 c), where deposition seems common for downward motion. Additionally, the deposition rate during PD does not show a strong dependence on the vertical velocity (not shown), although previous observations show a decrease of ice mass with downward motion (Shupe et al., 2008), suggesting potentially less deposition. To add to the discussion, the WBF occurrence with vertical velocity, shows a similar signal as the deposition for the PN and PD (Fig. 7 b-d). Even though WBF is computed from the evaporation and deposition, the evaporation occurrence does not seem to vary strongly with the velocity (Fig. C2). In addition, we found that condensation consistently increases for PN and PD towards upwards velocities, as one would expect (Fig. C2). As theorized by Korolev (2008) the WBF process is to a certain extent expected for downwards velocities, which can explain some of the behaviour of the deposition tendency during the PD. At the same time, it is expected that for upwards velocities saturation with respect to ice and water will set in, causing condensation to increase and therefore preventing the WBF process. Further, Omanovic et al. (2024) found a more prevalent WBF process in downdrafts in low continental stratus clouds using ICON-LEM with the SB microphysics scheme, albeit with different aerosol concentrations. Such a relationship of the vertical velocity to the WBF process we only found partially.

Reasons for the behaviours of the WBF process and deposition could be the lower number of frozen cases during PD or the differences in temperature range. Another reason could be differences in the vertical structure of the boundary layer and potentially increased moisture in the lower layers due to the fjord by Ny-Ålesund. One must keep in mind that the vertical wind is very narrowly distributed around $0 \text{ m s}^{-1}$, and only a few absolute high values exist. To study this further, other types of clouds that have stronger vertical velocities by nature are more suitable. For the liquid mass, similarities exist between evaporation and sublimation, as both show a decrease with upward velocity (Fig. C2 a and b). Condensation resembles the deposition more strongly and increases with upward motion (Fig. C2 c and d). The CCN activation is, by definition, dependent on the vertical velocity. Therefore, the increase with higher upward velocity, which we found, is as expected.

The last thermodynamic variable is the saturation with respect to ice and water. Processes that change the frozen mass are evaluated against the saturation with respect to ice. Here, riming stood out again. It can be seen in Fig. 6 c and d that riming increases with saturation in both occurrence and mean tendency. This fits the increase of riming found for higher upward velocities, as the saturation of rising air can increase. Some processes depend, by definition, on the saturation, for

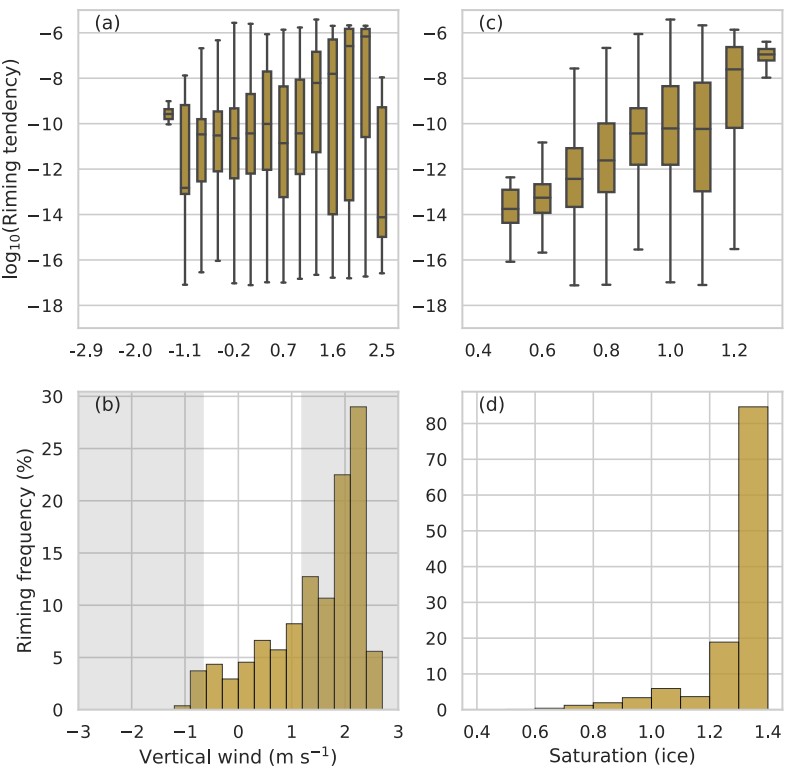

**Figure 6.** Dependence of riming on the vertical wind speed (left column) and the saturation with respect to ice (right column) for the polar night. The upper rows show the distribution of the riming tendency per bin as boxplots. The lower row shows the frequency of occurrence per bin. Bins of 0.1 are used for the saturation and 0.3 m s$^{-1}$ for the vertical wind. The grey shaded areas indicate each the lowest and highest 1 % vertical wind speeds.

instance, deposition/sublimation and condensation/evaporation. The signal found for these processes indicated evaporation and sublimation below saturation and condensation and deposition above saturation, which was, therefore, expected (not shown).

What was noticeable, though, was that the tendency of condensation showed a higher mean in combination with a much smaller spread in comparison to evaporation. It was generally an interesting finding that some processes, such as condensation and rain freezing, showed much less spread in their tendency than other processes in respect to all thermodynamic variables evaluated here.

## 4 Discussion and conclusions

This study evaluates the microphysical processes of the two-moment microphysics scheme from Seifert and Beheng (2006) as it is implemented in ICON. The area of focus is Svalbard, and only low-level clouds are selected for the analysis. A further separation is made between single-phase and mixed-phase clouds. Using a wrapper to run the microphysics scheme offline as

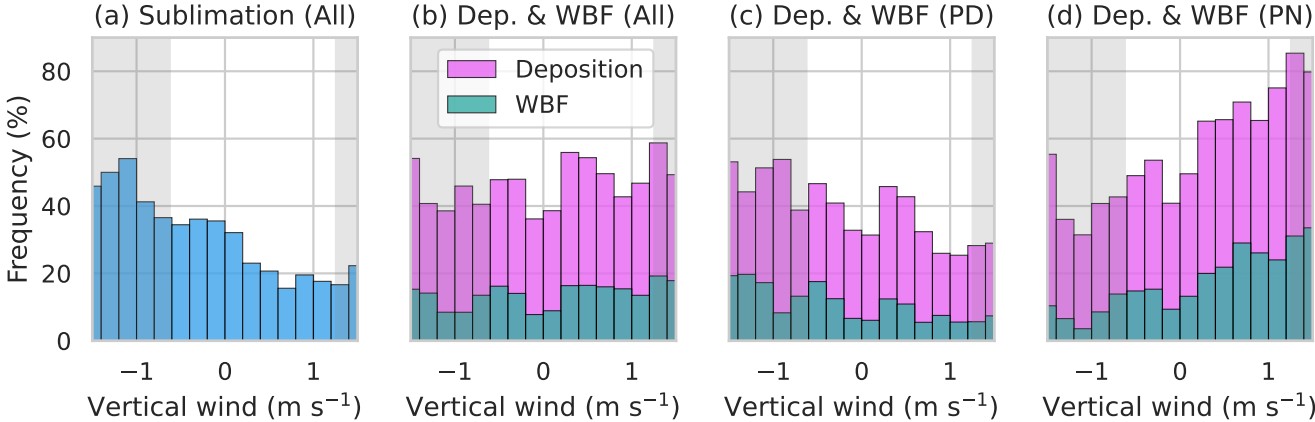

**Figure 7.** Occurrence of sublimation (a) and deposition and WBF (b-d) with vertical velocity (bin width: 0.2 m s$^{-1}$). Sublimation is shown for PD and PN (a), deposition and WBF for PD and PN (b), the PD (c), and PN (d). The grey boxes mark the lowest and highest 1 % of the vertical wind.

a diagnostic tool, the process rates per timestep are written out. In total, eight months, four polar night and four polar day, are simulated. This extensive dataset of microphysical process rates (Kiszler et al., 2023b) presents a valuable and novel resource

for further research, offering detailed insights and potential for addressing current knowledge gaps. The goal of this study was to use the created data set and determine which processes play the largest role in the phase transitions in Arctic low-level clouds and in what way they depend on temperature, vertical velocity and saturation.

It was found that the dominating processes in MPCs are phase transitions between liquid hydrometeors and vapour, as well

as frozen hydrometeors and vapour. The results suggest one possible approach to improving the representation of the phase-partitioning in low-level mixed-phase clouds in the Arctic could be to adjust the processes of evaporation/condensation and deposition/sublimation. Another approach could be to increase the activity of processes which are currently less active, as these may not be active enough. Further, the differences between polar night and polar day showed the importance of evaluating a large dataset covering different thermodynamic conditions. For instance, rain freezing seemed to be more important during the

polar day than during the polar night, while riming seemed to be more important during polar night. It is worth mentioning that nucleation processes only minimally change the mass directly, but the numbers of activated CCN and INP have an impact on other process rates. Fan et al. (2017) showed this in a case study for orographic clouds where evaporation became stronger than condensation for higher aerosol concentrations, whereas for lower concentrations, the process rates were similar. Therefore, although changes to evaporation/condensation and deposition/sublimation likely will cause large hydrometeor mass changes,

the interactions between processes also play an important role.

One such process interaction is the Wegener-Bergeron-Findeisen process, where liquid water evaporates and then deposits on ice due to the lower saturation of ice below 0 °C. To evaluate the WBF process, we selected cases where deposition and evaporation occurred simultaneously and used the minimum rate as approximation for the WBF tendency. We found a very frequent occurrence of 42 % of the WBF process in MPCs. Further, it seems like the deposition tendency determines the rate of the WBF process. Additionally, the evaporation tendency was evaluated in combination with the frozen and liquid mass. This showed a one-order-of-magnitude increase of the average evaporation, causing a significant decrease of the liquid mass. Combined, the results showed that a significant amount of mass follows the WBF pathway from liquid to vapour to the frozen phase with an average WBF tendency similar to the average deposition tendency. Reducing the WBF rate by reducing the deposition tendency may be a way to reduce the underestimation of liquid-containing clouds found in a previous study (Kiszler et al., 2023a). The finding, that evaporation increases substantially more than deposition was partially attributed to the implementation of the microphysical processes which favours excess evaporation when WBF is active. Also, the thermodynamics of WBF is expected to cause additional evaporation, but it is not possible at present to quantify this effect. Nonetheless, it is suggested that the tools and methods developed in this study can help making quantitative analysis of such effects and uncover the intricate relationships among moisture, temperature and cloud particle properties that affect the WBF process in numerical models.

We further explored how each process behaves in different thermodynamical regimes. Temperature is one important factor that determines the importance of a process. For instance, Fan et al. (2017) found that for warm orographic MPCs riming was similarly important for the snow formation as deposition, but for cold orographic MPCs deposition was clearly more important. In this study, differences are visible between the polar night and day. We found that melting and rain freezing play a larger role during the polar day while riming decreases in importance during that time. This dependence on temperature was further evaluated, and it could be seen that processes that change the mass phase between liquid and frozen show a stronger temperature dependence than those involving vapour phase transitions. The strongest temperature dependence was visible for rain freezing, which showed an increasing occurrence with increasing temperatures, while the mean frozen rain mass decreased. Interestingly, the distribution of riming for both occurrence and mean mass change in different temperature regimes was bi-modal, showing a minimum between -20 and -10 °C. The connections between the process rates and the vapour saturation and vertical wind were not as clear. This can partially be attributed to the narrow range of values for these thermodynamic variables given in low-level Arctic clouds. The clearest signal, in this respect, was the increase of riming with increasing upward velocities. This could be connected to the larger production of liquid in updrafts where condensation is more active. Another finding was that the WBF process did not fully behave with vertical velocity as one may expect it from the theoretical understanding (Korolev, 2008) and from another ICON-LEM case study (Omanovic et al., 2024). We found that for upwards velocities the WBF process seems to increase its activity during the PN.

As stated above, there are limitations to our approach, which might make it less insightful for cases where advection dominates the cloud hydrometeor composition, for instance, in deep convective cases. Nevertheless, when focusing on low-level

clouds in the Arctic, this approach provides valuable insights in regard to the processes inside the clouds, as demonstrated in this study. Additionally, the regimes of vertical velocity and temperatures studied here are limited to those of low-level clouds in the Arctic. Specifically over Ny-Ålesund which, as we found, represents more the decaying phase of clouds than the forma-
tion phase. Therefore, to create a broader picture of the microphysical processes in other cloud types, further studies, including stronger vertical velocities and larger temperature ranges, are necessary. This could, for instance, substantiate our findings in regard to the increase of riming with higher upward vertical velocities.

There are further factors that impact the process rates, as mentioned in the introduction. These include aerosol concentrations,
which can strongly impact the hydrometeor composition and cloud lifetime (Kalesse et al., 2016; Eirund et al., 2019). In this study, the CCN and INP are treated as maritime, which is more accurate than the default continental setting in ICON but still not completely correct. CCNs and INPs are another large area of active research, which is why this study focuses on the processes independently of aerosol influences. An interesting study would be, though, how tweaked CCN and IN settings impact the process rates using the approach presented here. Using the process rates and looking into the regimes where different processes
occur has shown that this method is also valuable for studying individual processes in greater depth. Being able to quickly change a process setting and get an idea of what might change in the model has proven easy and reliable. This encourages continuing to use tools, such as this wrapper, which simplify the untangling of complex cloud microphysics schemes.

*Code and data availability.* The microphysical wrapper code is stored on the DKRZ Gitlab. In the form used here, the wrapper includes ICON code which is licensed and the code is therefore only available on request. The ICON model code which we used is available for
institutions or individuals under a licence but a recently published open source version is available here: https://icon-model.org/. The low-level process rates, cloud selection and meteograms are available on Zenodo with the DOI: 10.5281/zenodo.10117706 (Kiszler et al., 2023b). A GitHub repository containing the code necessary to reproduce the results can be found here: 10.5281/zenodo.10945484

**Appendix A: Flowchart of microphysical wrapper**

**Appendix B: WBF dependence on SLF**

The supercooled liquid fraction (SLF) is computed based on (Komurcu et al., 2014).

$$SLF = \frac{r_{\text{liquid water}}}{r_{\text{liquid water}} + r_{\text{ice}}}$$

In Fig. B1 three different microphysical processes are shown: deposition, evaporation and WBF. The SLF generally has two maxima in occurrence towards 0 and 1. This figure demonstrates how the WBF process is limited by deposition for high SLF and by evaporation in low SLF regimes.

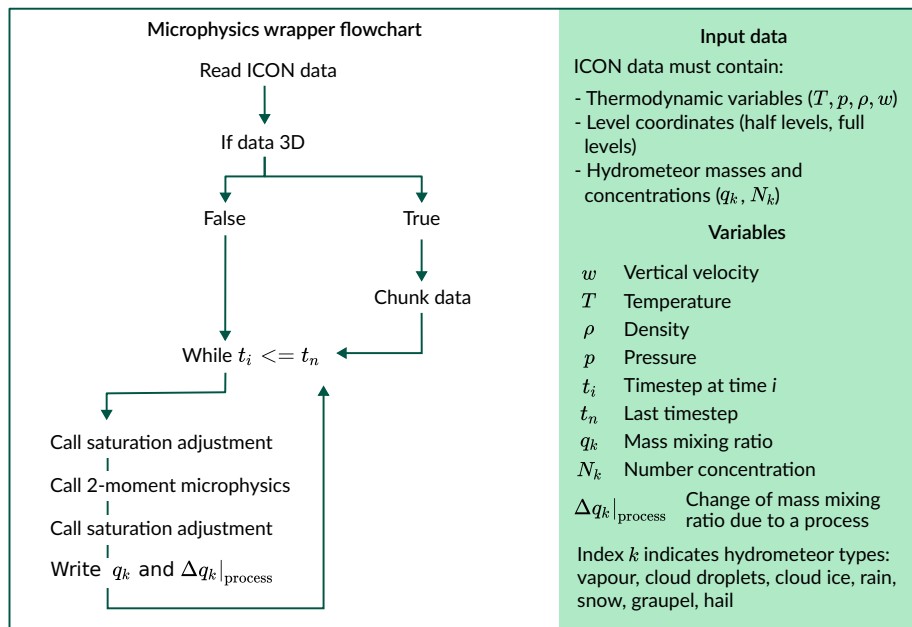

**Figure A1.** Flowchart illustrating the microphysical parameterisation wrapper. The variables and required input data are listed on the right. Any sub-selection of the data was done after running the wrapper. Each process has a separate tendency for each hydrometeor which it affects ($\Delta q_k|_{\mathrm{process}}$).

## Appendix C: Processes dependence on environmental conditions

This section provides additional figures for the dependence of the microphysical processes on the temperature, vertical velocity and saturation. They are complementary to the figures shown in section 3.3 and just show additional processes mentioned in the results.

*Author contributions.* TK carried out the wrapper implementation, data processing, and method development, created the visualisations and prepared the manuscript with contributions from all co-authors. VS and DO contributed to the conceptualisation, research supervision, and discussion of results.

*Competing interests.* The authors declare that no competing interests are present.

*Acknowledgements.* We gratefully acknowledge the funding by the Deutsche Forschungsgemeinschaft (DFG, German Research Foundation) – Project number 268020496 – TRR 172, within the Transregional Collaborative Research Center "ArctiC Amplification: Climate Relevant

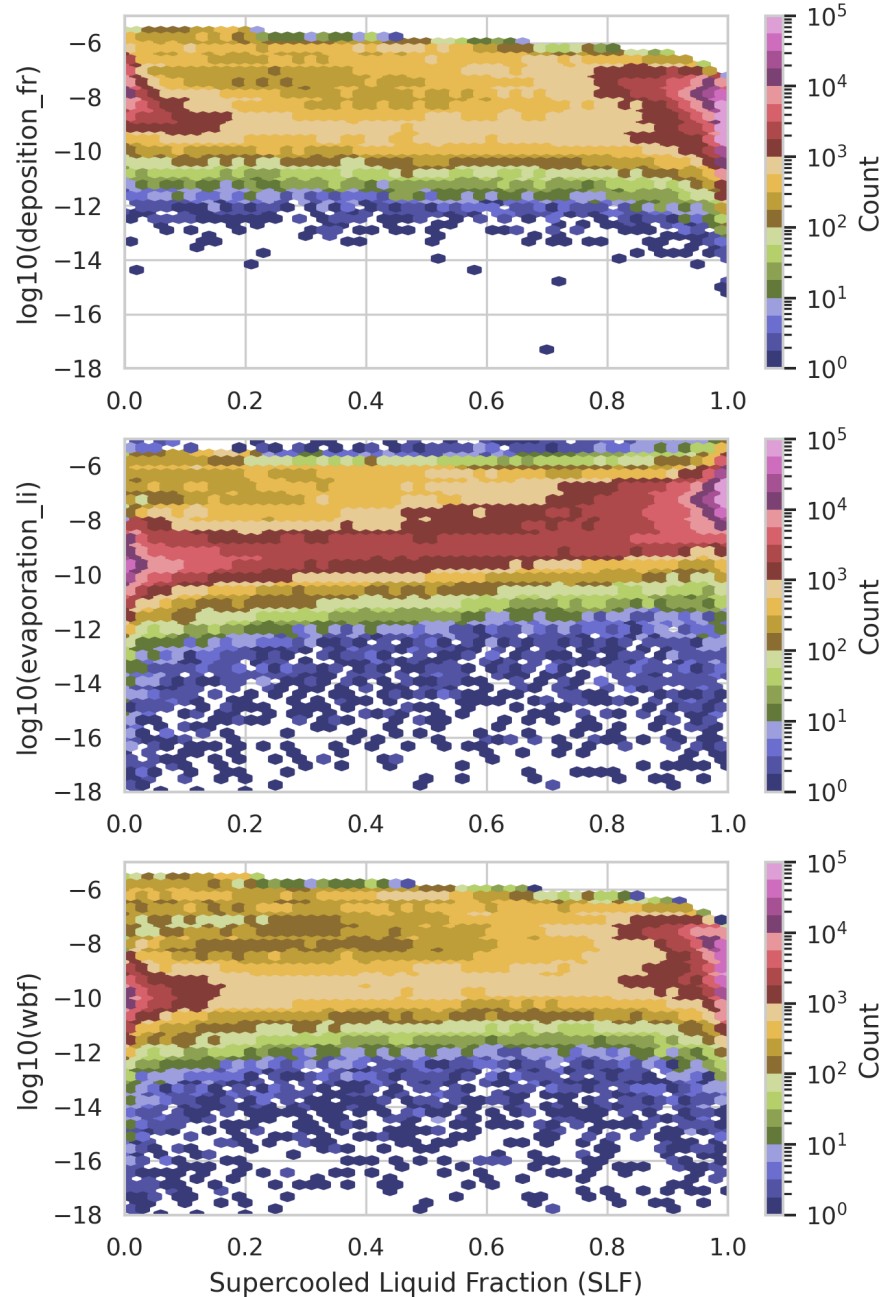

**Figure B1.** 2D-histograms demonstrating the connection between clouds with high SLF and low SLF and the microphysical processes: a) deposition, b) evaporation and c) WBF. The colourbars indicate the count of the cases.

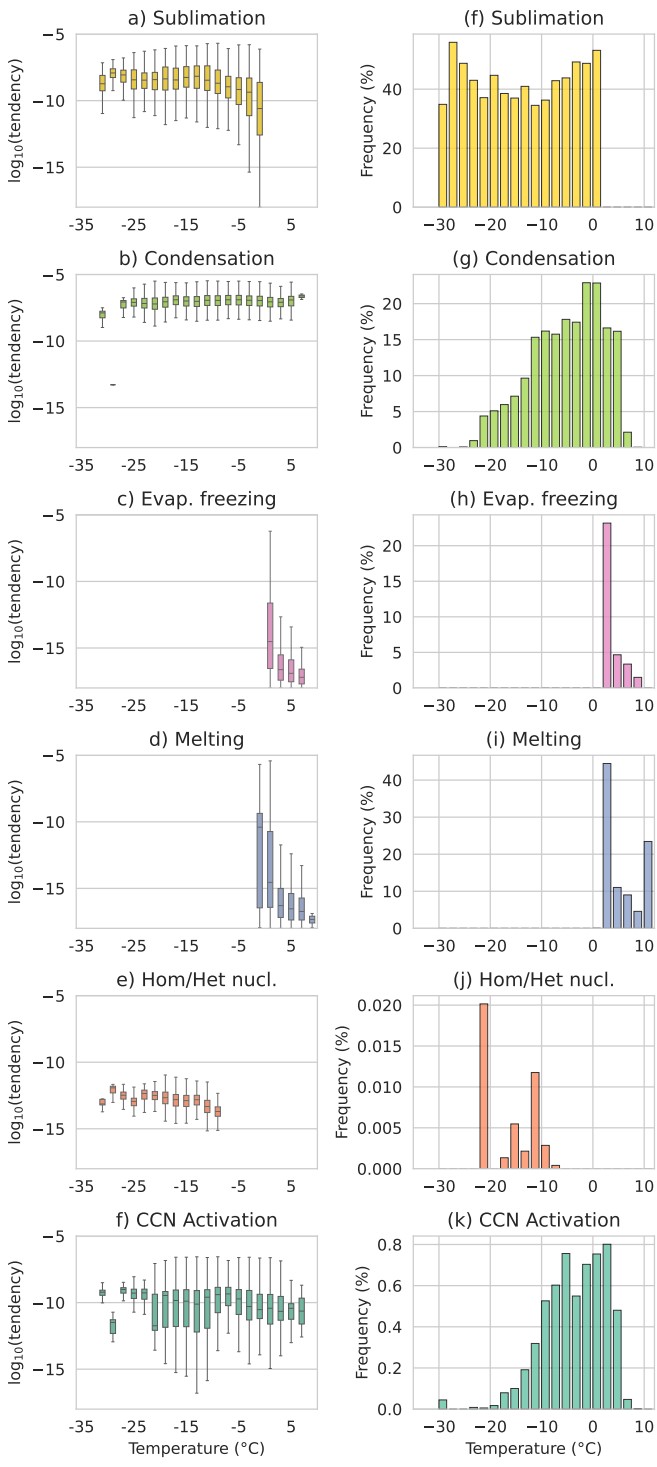

**Figure C1.** Temperature dependence of microphysical processes. Left column: box plots for temperature bins. Right column: occurrence for each temperature bin. The data includes the polar night and polar day, and bins of 2 °C are used.

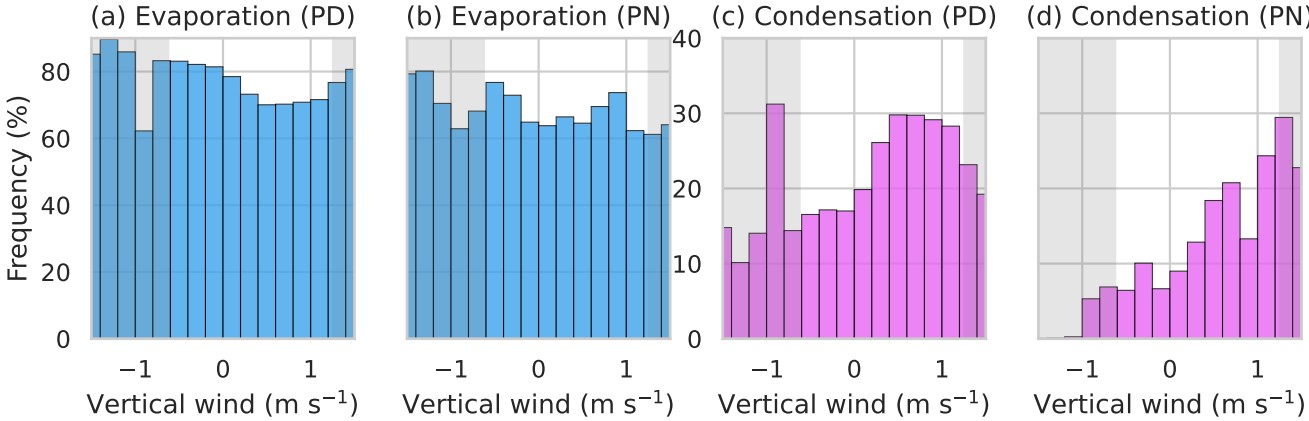

**Figure C2.** Dependence of evaporation (a and b) and condensation (c and d) on the vertical velocity. Split into PD (a and c) and PN (b and d) using 0.2 m s$^{-1}$ bins.

Atmospheric and SurfaCe Processes, and Feedback Mechanisms (AC)3. This work used resources of the Deutsches Klimarechenzentrum (DKRZ) granted by its Scientific Steering Committee (WLA) under project ID bb1086. We thank Axel Seifert for the support in implementing the microphysics wrapper. Further, the discussions with Rosa Gierens, Giovanni Chellini and Matthew Shupe helped shape the analysis. Two anonymous reviewers additionally helped to improve this manuscript, which we are grateful for.

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
