# Peer review of "Microphysical processes involving the vapour phase dominate in simulated low-level Arctic clouds"

_EGUsphere, 2023_

## Author Comment (AC1)

**Answers to the reviewers of the manuscript egusphere-2023-2986**

We thank the reviewers for the dedication of their time to review this manuscript. We appreciate their thoughts and willingness to engage in a discussion about the results of this study. We believe the helpful comments have substantially improved our manuscript.

**General summary of changes to the manuscript:**

Inspired by the comments provided by the reviewers, we decided to make some major changes to the structure of the content and add additional analysis. In the method section we have split the section 2.1 into 2 parts. One now solely includes the simulation description (2.1 Model simulations) and one includes a description of the selected data (2.2. Selected data). We moved the previous section 3.2 "Regimes of phases" into the now added section 2.2 "Selected data". This enables a clearer overview for the reader and makes them able to more easily find the description of what data is used. Therefore, some parts have be reformulated in the new section 2.2 in contrast to the original section 3.2.

Another change is the restructuring of the results. As the WBF process has received a lot of attention and further analysis was requested, we moved the WBF section "3.4 WBF in mixed-phase clouds" up, so that it is now the second results section 3.2. To add value to the manuscript we included another figure in the WBF section and expanded the discussion accordingly. Further, section 3.1 "dominating processes in low-level clouds" has been improved by splitting the results into different cloud types. For this, figure 1 (now figure 2) was completely revised. It still shows the importance of each process but distinguishes between mixed-phase clouds (MPCs) and pure ice or liquid clouds. The discussion was adjusted to match these changes.

Fig. 5 (previously fig.3) was shortened to only include the figures we perceived as most important and an additional figure in the appendix shows further processes. The WBF process was also added to Fig. 5. The discussion for this section has been adjusted accordingly.

A supplement was added to provide further figures. It includes a flowchart of the wrapper. As the number of mircophysical processes is so large we decided not to show all processes in the results section.

Line numbers refer to the revised manuscript where the changes are not marked.

**Reviewer #1 (bold is reviewer comment)**

**Overview and general recommendation:**

**This manuscript uses a new microphysics wrapper in the ICON-LEM model to extract microphysical tendencies related to low-level cloud phase in the Arctic. The authors wish to quantify the relative importance of different phase-changing processes. To simplify the analysis, all liquid and ice hydrometeors are binned together, respectively, so that only two categories are considered in the analysis. These tendencies are plotted as a function of temperature, vertical velocity, and ice saturation, and compared between polar day and polar night. The authors find that tendencies involving the vapor phase of water (e.g. evaporation, deposition, sublimation, and condensation) are especially important. The authors interpret the frequent occurrence of evaporation and deposition as evidence of the WBF process.**

**The approach used in this manuscript is novel and interesting. Instantaneous microphysical tendency fields are critical to mixed-phase cloud processes, but are rarely studied at this scale. The tool described and employed here gives granular insight into these processes in a valuable way. The writing is generally clear and I also find the manuscript to be well-organized. I find the results interesting, but their analysis and presentation leads to conclusions that feel vague and lack clear takeaways. Specifically, conclusions regarding the WBF process should explicitly quantify its importance as a function of other climate fields. I think that with some additional recommended analysis, the results will be more valuable to the broader community. As a result, I recommend major revisions. These concerns along with other more minor comments are included below.**

**Comments are formatted as:**

**Line number: "Text"**

**Specific Comment**

**1: "either"**

**Possible change: replace "either" with "both". "either" does not add to the information conveyed by the sentence since any bias must result from an over- or under-estimation of a model field.**

Thank you for the hint, this was changed to "both".

**6-8: "It was found that… vapour phase dominate."**

**The first conclusion is vague, what processes specifically vary and how so?**

**The second conclusion seems more important to the manuscript. Should it be listed first and highlighted?**

See answer for next comment, as larger changes were made addressing multiple comments.

**9-10: "Going a step further…occurrence."**

**The meaning and importance of this sentence is unclear to me. Wouldn't we effect the WBF effect to occur in the mixed-phase regime and to be characterized by liquid evaporation and ice deposition? Is new understanding enabled by this perspective?**

Answer combined to line 6-8 and 9-10: We agree that the original presentation of the conclusions did not highlight and clarify the main findings enough. We changed this by being more explicit in the formulation as well as moving the sentence on the WBF up before the dependence on the regimes (l. 7-10)

**31-32: Citation of McGraw (2023) and Shaw (2022)**

**I would highlight that these studies adjust both the ice nucleation and WBF processes while investigating the cloud forcing, not just the WBF process.**

**McGraw shows that agreement with observations can be achieved with multiple nucleation schemes if WBF is adjusted, while Shaw shows that multiple configurations of the WBF process rate and nucleation rates can be used to achieve good cloud phase agreement with observations.**

We thank the reviewer for commenting on how the sentences are formulated which include these citations. These citations are aimed at giving some background on the current interest and research in respect to the WBF processes. We added that the ice nucleation as well as the WBF were evaluated. l. 33

**33: "new: Classical Nucleation Theory"**

**This citation format is not familiar to me. Please review.**

It is helpful that reviewer #1 pointed out this cause for confusion. We originally had added information on the new setup used in that study but removed this now as it seems to distract from the main message of this sentence. This is meant as an example for a case where a model overestimates the liquid amount.

**53: "making the location…clouds."**

**Given later comments about the importance of surface type, can you comment on representativeness of Svalbard to the Arctic as a whole?**

The representativeness of Svalbard is indeed a topic worth discussing and ongoing work of colleagues at our institute is looking into this. Generally, one can say that these low-level clouds are a feature observed all over the Arctic. The processes occurring in these clouds, e.g. downdrafts/updrafts, composition, liquid amount, etc… have been observed during campaigns in other Arctic locations and are similar to those in Svalbard. A discussion on this is given in Gierens et al. (2020, https://doi.org/10.5194/acp-20-3459-2020). The big difference is here that we are dealing with orography and have little sea ice directly at this location. This impacts the boundary layer structure as for instance the air over the fjord is generally more humid than over the land (this is for instance shown in the cited paper Kiszler et al. 2023). At the same time we are focusing on microphysical processes and are being clear that what we find is valid for the conditions at Ny-Alesund. This is definitely to an extent transferable to other locations although the statistics would probably look different in respect to the lifetime of a cloud. We believe it would be valuable to continue this research and to look at several Arctic sites but that would go beyond our current study.

**53: "ICON-LEM"**

**Please introduce this acronym.**

This is now corrected in l: 56

**62 (and elsewhere): "approx."**

**This does not need to be abbreviated.**

This has been changed in all cases.

**64-65: "The forcing is…ICON-NWP runs."**

**For those unfamiliar with this model: Within the model domain, what boundary conditions are applied vs. determined by the model?**

**e.g. Greenhouse gas concentrations, surface temperature/type/fluxes, radiation fields at model boundaries.**

The ICON-LEM setup runs on a limited area which is forced using boundary and initial conditions from the ICON-NWP simulations (2.4 km resolution). This includes surface variables. Our limited

area ICON-NWP simulations are forced using global operational ICON-NWP simulations from the German Weather Service (13 km resolution). The simulation workflow is further described in the cited paper by Kiszler et al. (2023) where we evaluated the performance of our setup. To clarify the setup without the reader having to look it up, the following reformulation was made in l. 77 cont.:

"The initial and boundary conditions for each ICON-LEM limited area simulation are provided by an ICON-NWP simulation with 2.4~km resolution. This ICON-NWP simulation covers a larger domain and is forced by the operational German weather service global ICON-NWP runs."

**61-72 (entire paragraph)**

**Has this model/configuration been evaluated in its ability to capture MPCs and their radiative effects before? A simple, general overview of the model's performance would be useful here if available.**

**Convince the readers that this model is an appropriate tool for this study (e.g. fit for task). If the model has biases or limitations, how may they affect the conclusions of this study?**

The reviewer would find a short overview of the model performance beneficial. Such a model performance evaluation has been performed for this location by Schemann and Ebell (2020) and Kiszler et al. (2023). We agree that it would be helpful to summarize their findings. Therefore, we expanded this paragraph and added the following information which also adds to the understanding of why we are so strongly interested in the phase-partitioning:

l. 65 cont. "The general setup follows the papers by Kiszler et al. (2023) and Schemann and Ebell (2020) and a thorough evaluation of the model performance is provided in those studies. While the general performance of the model was found to be very good, there were some short comings. There it is shown that the cloud occurrence matches the observations well but that the occurrence of liquid containing clouds is underestimated by around 30%. "

**75: "10-8 kg/kg"; Later in the manuscript a threshold is described as 10^-18 kg/kg. Are these different thresholds or is it an error in the text?**

We thank the reviewer for pointing out this potential for confusion. These are two different thresholds, one for the hydrometeor mass and one for the hydrometeor mass change (process tendencies).  To avoid confusions the following sentence was added in l. 137 "This threshold is much lower than the threshold for the hydrometeor mass ($10^{-8}$~kg\;kg$^{-1}$) as the microphysical process tendencies, changing the mass, can be very small."

**89-94: Entire paragraph; How is the model vertical coordinate handled when identifying clouds and microphysical tendencies? Are outputs also produced on a vertical grid for cells where a cloud is present?**

The tendencies are computed for all grid cells on the same vertical grid as in the model. This is done independently of whether there is a cloud or not. The difference is that if there is no hydrometeor mass yet, and the saturation does not allow the formation of any, then the process tendencies will just be 0 (or very very small which is then the numerical noise which needs to be removed).

As the reviewer indicates an unclear point here the following information was added to the text in l. 122 "and on the same vertical grid as used in the model."

**96-97: "This approach…single processes.", Are the rates produced by this approach identical to those interactively seen by the model during the run?**

The reviewer points towards an important aspect of using a diagnostic tool instead of the direct output. In general the same functions run as in the model, so given that the input to the function is the same in both the model simulations as well as in the wrapper, the same result will be produced. The difference is that in the model simulations also other physics and dynamics run during one timestep. This means that the input to the microphysical functions will be slightly different in the model than in the wrapper as we do not run other processes during our timestep. This does not necessarily affect our results much as we know exactly what input we provide to the microphysical functions and we are not interested in the other dynamical processes during one timestep.

**98-101: "Another advantage…schemes have run."; This is a very important note. So the saved output is not merely a subset for MPCs, it is taken during a different part of the model updating loop? A flowchart figure showing this could be really helpful for a reader!**

The reviewer understandably has further questions regarding the way the wrapper works. To improve the clarity we followed the suggestion to add a flowchart. This has been added in the appendix (Appendix A, Fig. A1)

**112-114: "In this study…compensate each other."; Does this simplification create any potential shortcomings in the analysis? E.g. Phase partitioning is also dependent on sources and sinks of hydrometeors, so conversion of cloud ice to graupel/hail could quickly deplete ice and modify the phase partitioning.**

We appreciate the reviewer for taking the time to think through our methods and results. We agree that this could be a relevant aspect as an increase in fall velocity would diminish the amount of ice as stated if it precipitates. As we focused on the general ice and liquid mass using this simplification makes it clear how each phase is directly impacted. At the same time we do not explore the precipitation rate here and would rather suggest to create a separate study looking into different microphysical pathways. Such a study would rather take the form of the work by Barrett and Hoose (2023) and potentially one could include actual observations and forward simulations but that is not something we wanted to achieve here.

**128: "making the frozen mass increase due to riming smaller."; meaning of "smaller" is unclear.**

We thank the reviewer for making it clear that it is not necessarily understandable what is meant by the enhanced melting. To clarify this we changed the text in l. 155

"If T$>$0\;°C enhanced melting after riming will take place, making the frozen mass increase due to riming less as not all liquid will freeze onto the frozen hydrometeors."

**141-142: One can see…frozen hydrometeors."; A clear example could make the meaning of this statement more clear.**

This part has been changed substantially. The figure which is meant in this sentence has changed and as described in the general summary up top the results section related to this were rewritten. We hope the changes of section 3.1 make the results more clear now.

**148-149: During the polar day…less frequent."**

**The process rates seem to depend heavily on the mean state of the cloud phase. Does this present a chicken-egg problem? i.e. Is the cloud phase the result of the process rates or are the process rates the result of the cloud phase? This may be more of a question for the discussion, but I think it is important to address.**

**Also, do the rates generally balance each other? If sublimation decreases during the day does a vapor to condensate or ice transition also decrease to maintain balance?**

**If equilibrium is not achieved, is this the result of ignoring advection?**

**I guess what I am asking here is what does the budget for vapor, liquid, and ice look like using this analysis? I think that this would be a helpful way to visualize the balance of processes at play and also to understand processes that may be missing from this analysis (i.e. a not-closed budget implies important roles from advection, etc).**

We are grateful that the reviewer has thought about the meaning of these results. As mentioned before, this section has changed quite a bit but these question still remain relevant. In the manuscript l. 129  states that "One must keep in mind that any transportation (advection and precipitation) of hydrometeors cannot be included as the model itself is not run.". The reviewer therefore assumes correctly that the mass balance cannot be closed here by simply looking at the mircophysical process rates. What we have done to estimate how much advection plays a role is to compare the hydrometeor masses of each timestep between the wrapper and the model run. This showed us that generally below 10% of the mass change cannot be explained by the microphysical processes alone. The following figure 1 shows this as an example for a single case study.

[Figure]

Figure 1: Absolute difference of the wrapper and original simulation plotted against the original mass simulated by ICON-LEM. The hydrometeors are a) graupel, b) cloud droplets, c) rain, d) cloud ice and e) snow. The data includes 3 hours with a timestep of 3 seconds on 150 levels. The number of occurrences per bin is colour-coded, but as the ranges are different, the subplots use different colourbar ranges. The red dashed lines indicate the relative differences of 10% and 1%. (Taken from PhD thesis: Kiszler T. Improving our understanding of cloud phase-partitioning usning long-term cloud-resolving simulations of Svalbard. 2024)

**158: This is likely…is used here."**

**See question about the representativeness of results above. How well can the results be generalized for the entire Arctic if most is ocean? Could you repeat the analysis for an ocean gridcell to see if the conclusions change?**

This comment picks up the topic of a question before and we would like to refer to the answer of the question further up.

**Would you expect the conclusions to change with a different surface type?**

We would expect different conclusions to the point that in other locations the clouds may be in the process of forming in stead of decaying as they are often here. At the same time the physics will be the same and for a given thermodynamic conditions we would expect the same statistical behaviour. This would presumably be independent of the surface type. What may happen is that for instance the temperature, vertical velocity and saturation distribution would look different affecting potentially which processes dominate and therefore changing how fig. 2 would look. This would indeed be worth looking at in a study with a different focus as the interactions of the processes may also change.

**162-163: "while processes…not as relevant."**

**I would consider clarifying where the WBF process falls here. I assume that it is not considered a liquid to frozen process because in the model there is an intermediate vapor phase step. But in some models the WBF process directly shifts liquid water to ice, right?**

We appreciate the reviewers interest in understanding more about the WBF process. Therefore, we computed the WBF tendency using the deposition and evaporation and added it to the analysis (Fig. 2 WBF added as process). To improve the analysis we also split the cloud types to indicate what role each process plays in specific cloud types. Further, we expanded the section focusing on the WBF process by adding further analysis.

**165-166: To get an…the hydrometeors.”**

**Would you consider also looking at the process rates as a function of the vertical coordinate? E.g. height above cloud base or optical depth?**

**What about the vertical coordinate? Given the strong vertical phase partitioning of mixed-phase clouds, I might expect a very different balance of process rates between liquid cloudtops and the icier cloud interiors.**

We agree that this is an interesting question and from single cases we could see that there is a dependency on the location in the cloud. For instance evaporation occurs more often in lower areas where there is precipitation. As we focused on adding further analysis regarding the WBF and improving/refining the existing results, we have not added more results in this direction.

**177-180: “Of the total…31% contained ice.”**

**This is really interesting! Could you include histograms of liquid water content and ice water content for polar day and night as well?**

It is good to hear that the reviewer finds our results interesting. Considering that the length of the current revised manuscript already has increased quite a bit we have decided not to add further figures.

**201-202: “The homogeneous… further included.”**

**Are the statistics bad? I would be interested in seeing these values plotted if they are interpretable. As you previously noted, the mass tendencies from these processes are small, but the number tendencies are very important.**

Nucleation is indeed seldom but we show it now for the temperature dependence in the appendix in Fig. C1. To evaluate these processes it would make more sense to use the number concentrations as metric, which we do not have as a wrapper output currently and it would require a large effort to add this in the implementation of the wrapper and then rerun the processing. It would though, in future work, be worth investigating especially in the context of the aerosol impact.

**Figure 3:**

**Plots in the left column would be easier to read with more standard temperature increments (e.g. 5C) and ticks.**

This has been changed in the now figure 5.

**220-221: “Reasons for…temperature range.”**

**I think these are good hypotheses, especially given previous arguments that PD/PN differences result in part from differences in temperature and cloud phase. To test this hypothesis, you could plot 2-d histograms of deposition frequency and tendency as a function of both vertical velocity and temperature.**

This is a good idea and we explored this, but did not find a clear signal in the 2d histograms we made.

**223: "Ny-Å"**

**Abbreviation is not needed.**

This is corrected.

**225-226: "For the liquid mass…upward motion."**

**Could you include these described results as supplementary figures?**

We have modified this paragraph and included evaporation and condensation with vertical velocity in the appendix Fig. C2.

**240: "The introduction mentions that"**

**Remove this text.**

This paragraph was rephrased and that part removed.

**241-242: "As shown in…underestimated.";  This evaluation of ICON-LEM should be included in the introduction so readers have an understanding of the performance of the model and if it is fit for task.; Underestimated by how much? Can the model be trusted for this study? See fit-for-task comment in the methods.**

This comment connects to an earlier one and the changes mentioned there, are aimed at also addressing this question. To be more specific about the model performance we mentioned some results which we found especially relevant here in l 68

"In Kiszler et al. (2023) it is shown that the cloud occurrence matches the observations well but that the occurrence of liquid containing clouds is underestimated by around 30%."

**244-246: "The hypothesis…phase cloud.";  The wording here is unclear to me. The WBF process is active when the air is saturated with respect to ice but not water, so condensation does not occur. Evaluating the role of the WBF process requires knowledge of the temperature, right?**

We thank the reviewer for indicating potentially confusing wording. We have changed the paragraph to make it more clear what we are trying to look at and what we expect to see. (l. 199 to 204)

**Figure 4: Recommendation for readability: Updates x-ticks in subplot (a) to be centered around zero. Matching panel b would be easiest for readers.**

The figure (a) is currently centered around 0 and it is due to the way python seaborn boxplot and matplotlib deal with the xticks that it looks different. Both a and b use the same xticks range and

binning so it is unfortunate that the x-axis turns out differently. After some experimenting we decided to leave it as it is even though it is slightly suboptimal.

**247-249: "was produced…deposition occurs."; What is the relative frequency of each set? i.e. How often is this simple definition of the WBF process happening when both cloud phases are present?**

This is a great question and we have added the occurrence rate of the WBF process in Fig. 2 to identify the relevance of the WBF process. It occurs roughly 42% of the time in MPCs.

**261: "For both…statistically significant."; Can you state what method of significance testing was used?**

We use the Kruskal-Wallis test as implemented in Python scipy. Information added in l. 221 and the code can be found in the published jupyter notebook: "fig_03_04_wbf.ipynb"

**263: "one-order-of-magnitude"; This is a fourfold increase, not quite an order of magnitude.**

This was indeed not perfectly formulated and we have changed this in l. 222-223 to:

"When both processes occur at the same time, the average deposition rate experiences a fourfold increase ($9.8 \cdot 10^{-9}$ to $3.9 \cdot 10^{-8}$ kg kg$^{-1}$), while the average evaporation rate also increases by around one-order-of-magnitude ($2 \cdot 10^{-8}$ to $1.3 \cdot 10^{-7}$ kg kg$^{-1}$)."

**263-264: "increase…same order of magnitude"; If the absolute increase in evaporation is not matched by the increase in deposition, are there other important terms in the moisture budget?**

Yes, not all vapour may be transferred to the ice state. It could be a question of advection and the saturation adjustment is called before and after the deposition routine while the warm microphysics are only called after the deposition, these can all impact the amount of vapour available for evaporation. Therefore, it is possible that more evaporation takes place than what deposits in that timestep.

**240-267: Section on the WBF process.**

**I think that there is an excellent opportunity here to explicitly quantify the importance of the WBF process and evaluate it as a function of the variables used previously (vertical velocity, temperature, etc). Similar to the methods used to separate the different cases, you could calculate a simple estimate of liquid-to-ice flux due to the WBF process by taking the minimum of the evaporation and deposition tendencies when temperature is between 0 and -38C (or using another approach if one is more valid). This new WBF tendency could be compared with the others (as in figure 1) and also plotted as a function of vertical wind speed, temperature, and saturation. Additionally, you could compare this WBF tendency to the sum of all liquid-to-ice tendencies to see what fraction of liquid-to-ice transitions can be attributed to the WBF process and how that fraction changes as a function of vertical wind speed, temperature, and saturation. I think that this would allow you to quantify your results in a clear way for the audience.**

**In summary, I recommend:**

- **Calculating a new WBF tendency as described above.**
- **Including that tendency in Figure 1.**

- **Plotting both the WBF tendency and the fraction of liquid-to-ice flux it accounts for as a function of temperature, vertical wind speed, and saturation.**

The reviewer shows great interest in the WBF process therefore, we have expanded our analysis. We have computed the WBF tendency as suggested and added it in Fig. 2 (previously Fig. 1). Additionally, we have expanded the analysis to increase the understanding of the WBF process and to quantify better when it occurs and under what circumstances. This analysis could further be expanded but we believe the provided additional results add significant value to this study. Regarding the change of the liquid-to-ice fraction, it would be necessary to compute the ice and liquid mass directly before and after the evaporation and deposition and combine this. This is not so trivial because these functions are called at different times in the code and other processes are called in between. Therefore, one does not have the WBF process directly and cannot directly asses the changes of the mass fractions. This is the reason we added an evaluation of what SLF regimes it occurs in to see whether there was a dependency in this respect. We also added the WBF in Fig. 5 and 7.

**Figure 6:**

**Suggestions:**

**- Use consistent labelling of the sets in the figure legends.**

**- Check if colors in panel (a) are colorblind friendly.**

**- Report the fractional occurrence of each set in the figure label.**

We thank the reviewer for the suggestions and have considered them for the now Fig. 3. The labeling is created in a way that it is clear what data is included in the subsets. The reason why we do not just use "no WBF" and "WBF" is because subplot a) shows a different sub-selection than b) and c). This is also indicated by the different color selection. The colors had been put through an online colourblind friendly check prior to submission, and we try to use different symbols, transparencies and line types in all plots where multiple colors are used. The choice may still be faulty as also an online simulator can only go so far. If the reviewer finds the colours hard to distinguish we will change them and would appreciate a hint what colours might be a better choice as the online tool seems not to be sufficient in every case. Regarding the occurrence, this is a good thought and we added it in the text as well as in the figure caption.

**276-278: "The results suggest…active enough."; This conclusion feels pretty vague, could you be more specific? Do the conclusions of this study point to one approach versus another?**

This is indeed slightly vague and we have added a sentence in the WBF paragraph highlighting a potential next step to take ( l. 346 cont. "Reducing the WBF rate by reducing the deposition tendency may be a way to reduce the underestimation of liquid-containing clouds found in a previous study Kiszler et al. (2023)."). The issue here is that the microphysical processes interact with each other to an extent that it is very difficult to say what will happen if one changes one process equation. We have done some small tests changing the deposition rate and this impacts almost all other process rates as well as the masses in a non-linear way. Therefore, we want to be very careful about being more concrete in this respect.

**278-279: "Further…large dataset."; What does this say about the importance of the mean atmospheric state? How does ICON-LEM's representation of the mean state influence the conclusions of this work?**

The representation of the mean atmospheric state as well as boundary layer processes play an important role for the low-level clouds of the Arctic. This was also the reason why this model has been evaluated before being used to go into the details of the clouds.

**280-282: "It is worth…process rates."; So mass tendencies from nucleation processes are small but the number tendencies are important? Could this help guide model development and tuning? For example, could nucleation only influence ice and liquid number tendencies and not mass tendencies without a large effect? (just for low-level mixed-phase clouds?)**

We appreciate the thoughts of the reviewer in respect to this topic. To guide model development with respect to nucleation and activation we believe it would be necessary to also look into the number concentrations. The challenge here is that growth processes need nucleated or activated particles to begin with. Looking at this would for example be worthwhile if one can use different aerosol numbers. This would be worth setting up another study though where one can look at this in detail.

**305-306: "and the results suggest…frozen phase."; This language is too vague. How much of the mass tendency occurs via the WBF process?; A simple estimate of the WBF process could be calculated as minimum(liq-to-vap,vap-to-ice) for individual timesteps.**

The reviewer highlights that it would be helpful to be more explicit about the results. Therefore, we have rewritten this paragraph to a large extent to match the added results and clearly state the main findings and their potential implications in the summary. (l. 339 to 349)

**318-319: "In this study…completely correct."; I think this should be included in the introduction/model description. Are the aerosols prognostic, do they evolve in time?**

The aerosols are described in the model description in l. 74-77 "We use the Segal and Khain (2006) cloud condensation nuclei (CCN) activation with maritime aerosols, as well as the heterogeneous ice nucleation from Phillips et al. (2008) with the maritime aerosol concentrations.". As we do not look into aerosol effects, this should be sufficient for readers to find more information in case they want to investigate this topic further.

**Reviewer #2**

**This study presents an analysis of phase-change process rates in Arctic clouds above Ny-Alesund motivated by a need to better understand phase-partitioning in mixed-phase clouds. I think that analysis of process rates can be a valuable way to understand clouds and that is something that should be done more often. That said, I have some concerns about the applicability of their results to mixed-phase clouds generally. Even aside from these concerns, I'm not sure what the author's main conclusions are. The only specific conclusion in the abstract is that "the importance of a process varies for the polar night and polar day … phase changes that involve the vapour phase dominate." I'd argue that neither of these are particularly novel results. That said, I think that with some additional analysis, the paper could provide more insight than it does in its current form.**

**Major Comments:**

1. **Figure 1 shows that the liquid water budget is clearly not closed. The evaporation and condensation conditionally-averaged rates are essentially equal (my understanding is that averages are taken over all points that meet the minimum rate threshold), but evaporation is 4 – 8 times more frequent. That the budget is not closed is because the authors analyze rates through only a single grid column over Ny-Alesund. The results suggest then that clouds are advected over Ny-Alesund and not generally forming over Ny-Alesund. In short, Ny-Alesund does not capture the full life-cycle of clouds. This is a major limitation of the study, and a limitation that is not discussed by the authors. I find it very difficult to interpret the results without any sense for the underlying distribution of lifecycle stages of the clouds (and I realize that even quantifying the lifecycle is non-trivial). I think that this concern could be partly mitigated by focusing less on the tendencies in the dataset as a whole and more on subsets of data that include only clouds that meet certain criteria. For example, subset by clouds that are growing and clouds that are dissipating, or by clouds that are becoming more glaciated and clouds that are becoming less glaciated.**

We thank the reviewer for thinking through our results and have addressed this by discussing this more in depth in the second paragraph of the results section 3.1. Additionally, in the conclusions we have added this limitation explicitly. l. 370 " Specifically over Ny-Alesund which, as we found, represents more the decaying phase of clouds than the formation phase."

This said, we believe that the chosen data is still very valuable because many campaigns, long-term observations and model studies focus on Ny-Alesund. Additionally, something lacking in most discussions of the low-level clouds in this region are the process rates. This is where we can provide many meaningful insights and quantify the extent to which processes are active in this model. It is true that it would be beneficial to cover a larger spacial area, but as is known from any modeling study, then one cannot look at this high temporal resolution required for the analysis of microphysical process rates. In light of the current work done by others in and around Ny-Alesund, this study adds an important piece to the puzzle. Further, the microphysical wrapper has now been used in a first study and the next study could for instance cover a larger domain while reducing the time coverage.

1. **The authors are motivated by the need to understand phase partitioning, but aside from the WBF analysis toward the end of the manuscript, there is no explicit analysis of phase partitioning. Why not restrict analysis throughout the entire manuscript to mixed-phase clouds? And/or examine, say, tendencies of ice/liquid water fraction and identify the processes that are most important for changing this fraction? These processes may or may not be the same as the processes that are most important for the total liquid change and total ice change, if for example, two processes are well correlated and offsetting one another, or if for example, a process such as condensation is important only when a cloud is predominantly liquid. If the authors were to go down that route, I think it would be useful to additionally include sedimentation fluxes in and out of grid boxes since that is also a process that could change the local partitioning. It could also be interesting to examine the phase partitioning processes as a function of height in the cloud – perhaps phase-partitioning processes important near cloud top are not important in the precipitating regions of the cloud and vice versa.**

We appreciate the critical feedback and have addressed this by adjusting our analysis to directly differentiate between mixed-phase and single-phase clouds from the beginning of the results. One must add that any mixed-phase process such as riming or rain freezing will by default only occur in mixed-phase clouds so already in the first version such analysis was included. We have made this distinction clearer now and have further added insights into the WBF process. This led us to redo fig. 2 and rewrite large parts of the WBF results, adding further findings (also in fig. 5 and 7). We see that the vertical distributions in the clouds are an interesting point but have focused on adding value to the existing results. In fig. 4 one can already get an idea though as we used the temperature distribution to evaluate the WBF in connection to other processes. In respect of the liquid-ice fraction we are limited by the output and microphysical implementation in the ability to exactly say what the change to the fraction was after each process. That is mainly because the saturation adjustment is called twice, once before and once after the warm and cold microphysics. This makes it difficult to retrace at what point the ice-liquid fraction changed as we had to find out while trying to evaluate this. As this would indeed be interesting to see though, we would suggest adding this as output for each process in the wrapper. This could be achieved in a succeeding study.

**In short, I think that the analysis could have been more creative to mitigate weaknesses in the data sampling and to provide more insight into phase partitioning. I don't think the authors need to take all of my suggestions, but I think that major revisions could address these weaknesses and produce a study that more directly addresses the gap in knowledge that they identify and that will be ultimately more impactful for the community.**

**Minor Comments:**

**Line 45: I was uncertain whether "deposition rate" here referred to the deposition of snow to the ground surface, or the deposition of water vapour to the crystal surface. I think now that it is the former.**

In Kalesse et al. (2016) they study the microphysical process of snow deposition rate from the WRF model as the depletion of water vapour can determine the amount of vapour which can form liquid droplets. They compute it "[…] averaged between cloud base and 200 m below cloud base where snow growth rates are largest.".

**Line 98-101: I don't understand this advantage. Are you writing the thermodynamic variables as they exist immediately before the call to microphysics? If so, I don't believe that this was stated explicitly.**

We thank the reviewer for pointing out that this sentence was confusing. We have removed this and the following sentence because it causes more confusion than explaining anything. We created a flowchart (Appendix A1) which hopefully helps understanding the overall way the wrapper works and what goes in and what comes out.

**Line 128: confusing wording, please rephrase**

We have changed this in l. 155 to: "In SB in ICON, this also includes the Hallet-Mossop secondary ice production. If $T > 0\;°C$ enhanced melting after riming will take place, making the frozen mass increase due to riming less as not all liquid will freeze onto the frozen hydrometeors."

**Line 195: Is "increases" supposed to be "decreases"?**

Yes, this is corrected.

**Lines 200-204: Most of these results are not shown, is that correct? If so, please say so explicitly.**

These results were indeed not shown but we decided to add them in the appendix (fig. C1) and to refer there so that the reader has the full overview. This section was also changed to incorporate the WBF process in fig. 5. and therefore the paragraph was rewritten in parts.

**Lines 234-238: Are any of these results shown?**

These results are not shown so we have indicated that in the text.

**Line 249: Presumably "no deposition occurs" is the same thing as "sublimation occurs" since we are only examining mixed-phase clouds?**

Yes, the formulation is just aimed to make the difference between the two sets clear. This paragraph was rewritten to make the wording and section content clearer.

**Additional changes by the authors:**

Code and data availability statement:

- Model code: The ICON code has become open source in January 2024, therefore we updated the link and added a short sentence mentioning this.

- Analysis code: The data analysis code which was used for the resubmitted manuscript version can be found under the doi 10.5281/zenodo.10945484

- Analysed data: The data has not changed so the same data set is still valid but an additional data formating script has been added in the code and can be used to store the data selection as netcdf instead of csv.

To be more consistent we now everywhere abbreviate the polar night and polar day with PN and PD, respectively.

The temperature, vertical velocity and saturation are not abbreviated by their symbols anymore (T, w, $S_i$/$S_w$), as these symbols where not really used.

---

## Referee Report (RR1)

**Revision 1:**

Recommendation: Accept with minor revisions.

**Comments to Author(s):**

Manuscript Number: egusphere-2023-2986
Manuscript Title: Microphysical processes involving the vapour phase dominate in simulated low-level Arctic clouds
Authors: Theresa Kiszler et al.

Overview and general recommendation:

The authors have made significant improvements to this manuscript. I find that nearly all my concerns have been addressed. My remaining comments are listed below. Beyond the minor corrections, my comments are largely concerned with how the authors discuss the imbalance between evaporation and deposition when the WBF process is active. Specifically, evaporation exceeds deposition, indicating a cloud state that is constantly losing water. The other reviewer raised a very similar point in their comment on the study's choice of location and the authors have partly addressed this in their response. However, I think that the results regarding the WBF process will be of greater value when discussed in the context of: a) the evaporation-deposition imbalance indicating a strong cloud water sink that may exceed the WBF mass rate itself, b) how the choice of location likely causes this behavior and the relevance of these results to Arctic regions characterized by high sea ice cover (as opposed to Svalbard).

Comments are formatted as:
Line number in trackchanges document: "*Text*"
Specific Comment

182-183: "Further…"
I think that a slightly longer explanation is needed here for readers to understand this. Additionally see later comments regarding the WBF values and why they sometimes exceed the deposition and evaporation values.

184-202: Whole paragraph.
The authors have done a great job introducing this important point early on. Given that the results presented here may depend strongly on location, I think it would be useful for the authors to discuss how location may affect their results in the discussion. Specifically: Is the result that the WBF process is deposition-limited specific to this case over land? Given that many studies of low-level Arctic mixed-phase clouds is often focused on their ability to persist over long time periods, is this case where at least 33% of the clouds are evaporating representative (line 234)? And if the evaporation exceeds deposition when the WBF process is active are these cases more representative of the WBF process sustaining clouds or cloud evaporation/glaciation? *I think that this kind of discussion will help readers understand how this study fits into the broader literature around Arctic mixed-phase clouds.*

249: "demonstrates visualizes"

Wording error here.

256-257: "This shows…WBF process."
Can you comment on the evaporation rate increasing more than the deposition rate (and that generally the evaporation rate exceeds the deposition rate)? Similar to a previous comment, the WBF process shifting liquid to ice is mostly discussed here but the high evaporation rate indicating instability seems quite important as well.

273-276: "We hypothesize…set in."
Interesting! So there may be an indirect effect of riming/rain freezing/secondary ice processes that enhances the WBF effect at relatively high temperatures?

296: Figure 4
If the WBF process is taken as the minimum of deposition and evaporation, how can it exceed either of them in this figure and else? I may be misunderstanding so an explanation to the readers could be helpful here.

351: "This we only found partially."
Check wording here.

427: "simultaneaously"
Correct to "Simultaneously".

425-428: "When combining…tendency."
I struggled with understanding the authors' meaning here and recommend revising these sentences.

429-431: "Additionally,…liquid mass."
What caused the 10x increase in evaporation? I don't think that I agree that it is the occurrence of the WBF process because the increase in deposition is much less, right? Doesn't this indicate that the WBF process is already strongly limited by the deposition rate? Would decreasing the deposition rate as the authors recommend just lead to the cloud evaporating instead of transitioning to ice? I understand that there is additional complexity here (the air may saturate earlier), but in general if evaporation increases more than deposition when both are active I would expect that enhanced evaporation to dominate the cloud changes.

464-465: "Specifically…formation phase."
See previous comments. The importance of the study's location and regime of cloud decay is closely connected to the imbalance between evaporation and deposition seen in the process rates and WBF process analysis. I think that linking these concepts together is a critical aspect of this paper and should be included in the discussion.

---

## Author Response (AR2)

**Answers to the reviewers of the manuscript egusphere-2023-2986 (round 2)**

We thank the reviewers for taking the time to review this manuscript a second time. The provided comments and questions regarding our study and it's presentation are appreciated. Based on the suggestions by the reviewers and the editor, we have updated our manuscript and believe that the changes have further improved the manuscript.

**General remarks:**
Based on the feedback by the editor we have reformulated the last sections of the introduction and made some changes in the summary, to make it clearer what the relevance of this study is. Further, the biggest changes made, are aimed at clarifying one major discussion point which has been raised. This was the discussion of the WBF process and why the evaporation and deposition rates are not in equilibrium. To address this aspect, changes have been made in the method, results and discussion sections.

Line numbers refer to the revised manuscript where the changes are not marked.

**Reviewer 1:**
**Recommendation: Accept with minor revisions.**
**Comments to Author(s):**
**Manuscript Number: egusphere-2023-2986**
**Manuscript Title: Microphysical processes involving the vapour phase dominate in simulated low-level Arctic clouds**
**Authors: Theresa Kiszler et al.**
**Overview and general recommendation:**
**The authors have made significant improvements to this manuscript. I find that nearly all my concerns have been addressed. My remaining comments are listed below. Beyond the minor corrections, my comments are largely concerned with how the authors discuss the imbalance between evaporation and deposition when the WBF process is active. Specifically, evaporation exceeds deposition, indicating a cloud state that is constantly losing water. The other reviewer raised a very similar point in their comment on the study's choice of location and the authors have partly addressed this in their response. However, I think that the results regarding the WBF process will be of greater value when discussed in the context of: a) the evaporation-deposition imbalance indicating a strong cloud water sink that may exceed the WBF mass rate itself, b) how the choice of location likely causes this behavior and the relevance of these results to Arctic regions characterized by high sea ice cover (as opposed to Svalbard).**

**Comments are formatted as:**
**Line number in trackchanges document: "Text"**
**Specific Comment**

**182-183: "Further…"**
**I think that a slightly longer explanation is needed here for readers to understand this. Additionally see later comments regarding the WBF values and why they sometimes exceed the deposition and evaporation values.**
From the reviewers comments, it has become clear that more explanation is necessary on how exactly the evaporation and deposition are implemented in the model. This is a relevant detail, therefore we have added more information on this in l. 180 *"An additional process which is not directly implemented in SB, but is analysed in this study, is the WBF process. As evaporation and deposition are needed simultaneously for the WBF process, it is possible to use their rates to compute the WBF rate. During WBF events, the second call to the saturation adjustment happens*

*in an atmosphere that has been deprived from moisture due to deposition on ice and hence causes additional evaporation."*

**184-202: Whole paragraph.**
**The authors have done a great job introducing this important point early on. Given that the results presented here may depend strongly on location, I think it would be useful for the authors to discuss how location may affect their results in the discussion.**
We understand the interest in the location dependency. We want to highlight though that we do not have an overview of the process rates in other locations, as so far generally only single case studies have been looking into the process rates to the extent we have done it here. It would be very beneficial if more studies evaluating process rates directly and not by proxy (i.e. number and mass concentration changes) would exist. As this is not the case we mention the location dependency but do not debate this in detail.

**Specifically: Is the result that the WBF process is deposition-limited specific to this case over land?**
We cannot answer this as we do not look at other model columns.
**Given that many studies of low-level Arctic mixed-phase clouds is often focused on their ability to persist over long time periods, is this case where at least 33% of the clouds are evaporating representative (line 234)?**
Here it is worth highlighting, that we have looked at 8 months of data and therefore argue that our finding are representative for the simulated clouds above Ny-Alesund during the PN and PD. The finding of such frequent evaporation could explain why we have found too little liquid in simulated clouds at Ny-Alesund in comparison with observations. It is worth pointing out that we are not trying to show what happens in Ny-Alesund in reality but what happens in the model and give ideas why the simulated clouds could differ from the reality.
**And if the evaporation exceeds deposition when the WBF process is active are these cases more representative of the WBF process sustaining clouds or cloud evaporation/glaciation?**
If the WBF process is occurring this is a glaciation process. It is possible that excess evaporation is decreasing the cloud mass though at the same time. This imbalance between evaporation and deposition is discussed below.
**I think that this kind of discussion will help readers understand how this study fits into the broader literature around Arctic mixed-phase clouds.**

**249: "demonstrates visualizes"**
**Wording error here.**
Thank you, corrected.

**256-257: "This shows…WBF process."**
**Can you comment on the evaporation rate increasing more than the deposition rate (and that generally the evaporation rate exceeds the deposition rate)? Similar to a previous comment, the WBF process shifting liquid to ice is mostly discussed here but the high evaporation rate indicating instability seems quite important as well.**
This is indeed a relevant aspect where the implementation details help to understand what happens inside the model and as described above we have expanded the explanation of the implementation. The evaporation is computed by the saturation adjustment which is called twice, once before and once after the other microphysical processes are called. In addition to showing this is in the supplement figure A1, we added this also in the process description in line 175 *"The saturation adjustment is run once before and once after the other microphysical processes (see Fig. A1)"* Therefore, not all of the evaporation in each time step may be directly connected to deposition. Evaporation could occur twice while deposition can only occur once. To make this clear the

following sentence was added in l 246" *The difference in rate change could be connected to the microphysics implementation, where the saturation adjustment is called twice in contrast to the deposition, which is called only once.* ". This is something where there exists further study potential into what impact the numerical implementation may play and also which cloud conditions allow for varying relative process importance. This is because deposition and evaporation are ultimately the result of a complex interplay of air humidity, temperature, and the properties of ice particles and liquid drops. We suggested that also the latent heat release resulting from the net glaciation caused by WBF could cause additional evaporation. Although that goes beyond the scope of this paper we believe it is worth pointing out so we added a discussion on this in the results  l. 391 *"The finding, that evaporation increases substantially more than deposition was partially attributed to the implementation of the microphysical processes which favours excess evaporation when WBF is active. Also, the thermodynamics of WBF is expected to cause additional evaporation, but it is not possible at present to quantify this effect. Nonetheless, it is suggested that the tools and methods developed in this study can help making quantitative analysis of such effects and uncover the intricate relationships among moisture, temperature and cloud particle properties that affect the WBF process in numerical models.* "

**273-276: "We hypothesize…set in."**
**Interesting! So there may be an indirect effect of riming/rain freezing/secondary ice processes that enhances the WBF effect at relatively high temperatures?**
Yes, it is interesting that there seems to be a connection. It must be taken into account though that the cloud occurrence also plays a role in fig. 4 as we just normalized by temperature there. We have also looked into whether the fact that there are more clouds around -3°C is the cause for this peak but even when normalizing also with respect to cloud occurrence, we can see that deposition behaves differently during PD than during PN and does not show the continuous decrease which we would expect towards 0°C. We have mentioned this now in the text in line 167 , but have decided against changing the figure as the figure which is normalized with respect to temperature and cloud occurrence could easily be confusing for readers.

**296: Figure 4**
**If the WBF process is taken as the minimum of deposition and evaporation, how can it exceed either of them in this figure and else? I may be misunderstanding so an explanation to the readers could be helpful here.**
The reviewer points out, that the explanation is not fully clear. We agree with this and have changed the caption to make this clearer "[…] The process distributions are normalized with respect to temperature but not cloud occurrence.".
Each process distribution is normalized, hence its integral over the temperature range is 100%. So if the WBF process distribution is focused slightly more around a specific temperature range, the percentage of its occurrence will be higher than deposition and evaporation. The calculation can be found in the following jupyter notebooks in the mentioned github repository: /notebooks_figures/fig_03_04_wbf.ipynb and /required_modules/process_class.py

**351: "This we only found partially."**
**Check wording here.**
Sentence corrected, thank you.

**427: "simultaneaously"**
**Correct to "Simultaneously".**
Corrected, thank you.

**425-428: "When combining…tendency."**

**I struggled with understanding the authors' meaning here and recommend revising these sentences.**
These sentences were reformulated to improve the clarity. L 382 "One such process interaction is the Wegener-Bergeron-Findeisen process, where liquid water evaporates and then deposits on ice due to the lower saturation of ice below 0°C. To evaluate the WBF process, we selected cases where deposition and evaporation occurred simultaneously and used the minimum rate as approximation for the WBF tendency."

**429-431: "Additionally,…liquid mass."**
**What caused the 10x increase in evaporation? I don't think that I agree that it is the occurrence of the WBF process because the increase in deposition is much less, right? Doesn't this indicate that the WBF process is already strongly limited by the deposition rate? Would decreasing the deposition rate as the authors recommend just lead to the cloud evaporating instead of transitioning to ice? I understand that there is additional complexity here (the air may saturate earlier), but in general if evaporation increases more than deposition when both are active I would expect that enhanced evaporation to dominate the cloud changes.**
We appreciate the interest the reviewer has shown in this. To address these questions several adjustments in different places of the text have been made and are listed in the above two comments which also refer to this topic.

**464-465: "Specifically…formation phase."**
**See previous comments. The importance of the study's location and regime of cloud decay is closely connected to the imbalance between evaporation and deposition seen in the process rates and WBF process analysis. I think that linking these concepts together is a critical aspect of this paper and should be included in the discussion.**
It is indeed noticeable that the liquid clouds generally seem to be in decay. We agree that it is very important to discuss the location aspect as well as this evaporation-deposition imbalance. At the same time we would disagree in linking the WBF rates too strongly to the location. We believe the imbalance of evaporation and deposition in WBF cases could be caused by the implementation.  It would require further research into different locations and cloud types in our view to expand the picture. Such progress is currently being made (for example Omanovic et al., 2024).

**Reviewer 2:**

**I appreciate the large effort that the authors put into revising their manuscript. I certainly think that it reads better now. That said, I do still have concerns about the discussion and the utility of the results.**

**Major Comments**
**1. A main conclusion that appears in the abstract and the conclusions is that "the dominating processes are phase transitions between liquid hydrometeors and vapour, as well as frozen hydrometeors and vapour." I agree that the results showed this, but isn't this just as we would have expected? For these clouds where presumably precipitation is typically minimal, the only way to make or destroy a cloud is through phase transitions with vapor. Unless all clouds were created as liquid and froze to make fully ice clouds (in which case, vapor transitions would be equally important to liquid-solid transitions), vapor transitions will by definition be more important than liquid-solid transitions.**
We thank the reviewer for their critical view on this and agree that this may sound like a basic finding. We would like to highlight though that for MPC it is not clear which processes exactly create the phase-partitioning. In this study we put numbers to the expectations and we show for example that although one may expect riming for instance to be very important, that this is not the

case here. This study takes our current understanding of Arctic low-level clouds and looks at how their processes are implemented and digs deep into the details of what exactly is happening. Using the process rates of the model, we show what happens and in addition link this to the lack of simulated liquid containing clouds in the model.

**2. In lines 326-330, the authors try to make suggestions for improving phase partitioning in models. It's hard for me to see how these suggestions follow from the results and to be honest the suggestions mostly boil down to "try everything." Since the results themselves are model-based and we have no comparison to observations, it seems difficult to say what the problem is with models (presumably the authors mean global models at much coarser resolution – this should be explicit). I think a better and more useful discussion would be about how others could use your results to evaluate their models and identify weaknesses.**

The reviewer wishes for more detailed suggestions about how to improve the model. In the discussion we provide several suggestions of what can be addressed as we know that the problem is the missing liquid water in Arctic clouds in ICON. We highlight that we believe the evaporation/condensation (saturation adjustment) and the deposition/sublimation are relevant targets to address. Additionally, we point out that potentially other processes (those which are less frequent) could be increased in efficiency. We cannot get more specific here because there are many unknown factors which are for instance the aerosol settings. The entire results of what we found can be used though to guide efforts in improving the representation of low-level clouds in ICON. The suggestions for specific process changes are based on our findings for each process rate.

**Minor Comments**
**ICON Simulations: Can the authors say a little more about the simulations? It says that each is run for 24 hours, but how many simulations are there? How are they distributed over the year?**

The simulation are semi-operational and run for every day (we added this in the manuscript). From this study we selected some data, as mentioned in section 2.2 "Selected data". As it states in Line 109 "Therefore, two sets of data are used. One covers the polar night (PN, November 2021 - February 2022) and one the polar day (PD, May-August 2021).". We then list how many days worth of low-level clouds we are evaluating. We believe adding another number to list the number of days the accumulated 8 months have in total will cause confusion because we do not analyse each day but only times with low-level clouds. Therefore, we will leave this as it is.

**WBF frequency: Have other studies tried to quantify the WBF frequency in clouds? If so, can you compare your results to these studies? If not, maybe this is something to highlight more explicitly as a novel contribution of this study?**

Other studies have tried to quantify the WBF process for instance using proxies such as the hydrometeor mass and number concentrations. This can be difficult to compare though as different locations, cloud types and timescales are evaluated. One paper which we have added now is by Omanovic et al. (2024) and complements our work. So far though, there have been no such detailed analysis of the process rates for a specific location over such a long period. This is where we can provide novel insights.

**Line 69: The authors state that the simulations underestimate liquid water by 30%. Can they comment on how this shortcoming impacts the analysis? It seems like this is a pretty large model bias for a study that is trying to understand phase partitioning better.**

We are glad that we made the main shortcoming of the model so clear. This said, this study was motivated in parts exactly by this underestimation of the liquid water. We want to understand what could be causing this, as ICON is not the only model that struggles with the representation of supercooled liquid. Therefore, this is not a shortcoming in our eyes but the reality of atmospheric models in their current state.

**Line 182: "The finding that all cloud types seem to be in the process of decay, where processes acting as sinks are dominating, is potentially a local feature as only the single column of Ny-Ålesund is used here." Why only "potentially" a local feature? I mean, it seems like it can't possibly be true that all clouds are decaying everywhere. Surely somewhere the clouds are forming.**

We agree that the clouds must form somewhere. The "potentially" refers to the aspect that a primarily decaying phase of clouds also could happen elsewhere. Further, and most importantly, we do not know exactly why this is what we found in Ny-Ålesund and therefore we want to be careful with our wording.

**Line 206: For clarity, specify that deposition and evaporation occur simultaneously. And in the following line, the second set is presumably evaporation with no deposition?**

Line 227 states "To evaluate this aspect the subselection of MPCs was evaluated where evaporation was occurring (75 % of MPC cases). This set was split into two sets. The WBF set consists of cases where deposition occurs simultaneously and where it is, therefore, sub-saturated with respect to water and saturated with respect to ice.". To us it is not clear how this could be more explicit as WBF is defined as evaporation and deposition at the same time. Further it states that "deposition occurs simultaneously" so in addition to evaporation, which is occurring in this data set as the sentence above states.

**Is it fair to say that this second set is where we have evaporation and sublimation? Likewise, does "no evaporation" mean "condensation"?**

Yes, "no evaporation" generally means condensation and "no deposition" means sublimation as we are not in an equilibrium where nothing would happen. As we use thresholds the cases of "no evaporation" or "no deposition" are defined as all cases which are below the given threshold of 1e-18.

**Line 226: We "found that the WBF process seems to correlate more strongly with deposition than with evaporation (Fig. 4)". Since the WBF rates is defined as the minimum of evaporation and deposition, this results says that the deposition rate is typically less than the evaporation rate. Doesn't this say that the combination of the two is acting to humidify the air and shut off the WBF process? This seems surprising to me since I tend to think of glaciation via the WBF mechanism as a runaway process rather than one that attempts to bring water back into equilibrium. Or does an imbalance in latent heating prevent this shutoff? I'd have to think through it a little more. I guess I'm generally wondering if more could be said about this result which seems potentially counterintuitive.**

The reviewer brings up an interesting thought here. We found that there seem to be two regimes where the WBF is limited by either deposition (which seems more common) and where it is limited by evaporation. These regimes are linked to the amount of liquid in the cloud and this can be understood better when looking at the supplement figure B1. It could be that there is a time-lag whereas the evaporation is faster than the deposition.

Another aspect to mention here is that the WBF process is not explicitly modeled in the cloud scheme. It rather results as a direct consequence of the implementation of the microphysics which allows the saturation adjustment to be called twice in each timestep in contrast to the deposition which is only called once. This causes the WBF to emerge as a way to restore the water equilibrium as the reviewer suggested. We realize that this was not so clear in the text and have added more explanation on this in l 180 *"An additional process which is not directly implemented in SB, but is analysed in this study, is the WBF process. As evaporation and deposition are needed simultaneously for the WBF process, it is possible to use their rates to compute the WBF rate. The saturation adjustment, which provides the evaporation rate, is computed twice in each timestep in contrast to all other microphysical processes. During WBF events, the second call to the saturation*

*adjustment happens in an atmosphere that has been deprived from moisture due to deposition on ice and hence causes additional evaporation."*
and explained how this may cause an imbalance between evaporation and deposition in l 244 *"The difference in rate change could be connected to the microphysics implementation, where the saturation adjustment is called twice in contrast to the deposition, which is called only once and the physics of the WBF process. Considering the typical thermodynamic situation characterizing WBF, the atmosphere is subsaturated with respect to water and supersaturated with respect to ice. This causes evaporation to occur during the first call of the saturation adjustment, providing more moisture to be deposited into ice as a result of the microphysics scheme, then during the second call of the saturation adjustment the atmosphere tends to return to the state it was before deposition happened. Because of this, intuitively evaporation would be higher than deposition."* Finally we discussed that the imbalance in latent heat should actually favor higher evaporation rates. *"Additionally, if deposition and evaporation tendencies would be the same there would be a net release of latent heat causing the equilibrium to shift towards additional evaporation."*

**Line 231: "As deposition should decrease with increasing temperature the peak at higher temperatures". It's not obvious to me why the deposition should decrease with increasing temperature, although I see later that you show this result explicitly. I suppose that supersaturation wrt ice tends to be higher at lower temperatures, but there's also less water vapor available. A quick reasoning for this statement could be helpful, especially for the reader who is not an ice microphysics expert.**
Deposition is implemented to be dependent on the temperature and saturation. In case of higher temperatures the required supersaturation wrt ice increases and therefore more moisture would be required to cause the same amount of deposition as at a lower temperature. As the reviewer observes, with increasing temperatures above -15°C, we found the deposition rate to decrease. We added that this statement is shown later to make the statement more clear: l 265 *"As deposition should decrease with increasing temperature (shown later), the peak at higher temperatures was not expected."*

**Line 264: "evaporation dominates throughout all temperature ranges where liquid occurs" Given that the mass budget is not closed, do the authors think this is a general conclusion or only one that is specific to this dataset?**
This is specific to the dataset. We cannot reliably generalize the conclusions of this analysis to the larger Arctic environment or even different climate regimes, but the developed tools can be easily used for data from different locations and/or time of the year, allowing to extend the study to diverse conditions.

**Line 295: "the WBF process is to a certain extent expected for downwards velocities" Why is it expected? Shouldn't downward velocities promote subsaturation with respect to both liquid and ice? Perhaps the authors just mean that the WBF process is possible in downdrafts?**
This is expected when the environmental conditions are favourable for the WBF process, which means for slight downwards velocities given the right saturation vapour pressure and temperature. Downwards velocities can promote subsaturation with respect to both liquid and ice but there are certain thresholds for this to occur. In the cited Korolev (2008) paper the theory behind this is elaborated as the critical vertical velocity depends on multiple factors and there are several regimes which can be differentiated.

**Line 343: "it seems like the deposition tendency drives the occurrence of the WBF process" I'm not exactly sure what this statement means. The authors showed that the WBF rate reflected the deposition rate, but since the WBF rate is the minimum of deposition and evaporation, doesn't that mean the evaporation rates are higher? I'm not trying to say that**

**evaporation drives the WBF, just that I'm not sure that the higher correlation with deposition necessarily implies anything about what controls the WBF occurrence.**

The reviewer pointed out that this sentence is not fully clear and we have revised it to capture better what we have found. L 385 *"Further, it seems like the deposition tendency determines the rate of the WBF process."*

**Technical Comments**

**Line 92: kg kg-1 formatting**
Thank you, changed.

**Line 194-196: Run-on sentence that is confusing.**
This was indeed a confusing end of the paragraph and we have improved it in l 217:
*"Evaluating this single column, shows that microphysical processes vary strongly in their importance and depend on the location studied. It is evident that the microphysical sinks found for liquid clouds are much weaker for mixed-phase and ice clouds. Especially, for the MPCs it became clear that the WBF process acts strongly upon the liquid mass and it is therefore worth further investigating its behaviour."*

**Further minor changes by the authors:**

l 20 wording: "… interest to which extent clouds play a role." → "... interest to which extent clouds play a role in this."

l 37 Corrected grammar: "… processes remains a challenge ..." -> "processes remain a challenge"

l 79 Missing lon-lat values "...diameter centred in Ny-Ålesund (Svalbard, lon-lat)..." → "...diameter centred in Ny-Ålesund (Svalbard, 78.9 °N, 11.9 °E)..."

l 114 Wording corrected: "In the analysis the temperature, vertical velocity, and ice/water saturation with dependency of different microphysical processes is discussed." → "In the analysis the influence of the temperature, vertical velocity, and ice/water saturation on different microphysical processes is discussed."

l 119 Wording corrected: "… both PN and PD, and both the PD and PN low variation ..." → " ...both PN and PD, and both the PD and PN show low variation..."

l 160 Correction: "… water vapour due to phase changes from frozen to vapour. "… water vapour due to phase changes between frozen water and water vapour."

l 226 Improved wording. "...whether the evaporation rate would increase due to the WBF process." → "...whether the evaporation rate increases due to the WBF process."

l 369 Increase clarity: "It was found that the dominating processes are phase transitions between liquid hydrometeors and vapour, as well as frozen hydrometeors and vapour." → "It was found that the dominating processes in MPCs are phase transitions between liquid hydrometeors and vapour, as well as frozen hydrometeors and vapour.

L 304 Improve clarity  and shorten sentence"… which potentially indicates that it may be more strongly influenced by other factors for negative temperatures, than other processes which depend

more clearly on the temperature. " → ". This potentially indicates that evaporation may be more strongly influenced by other factors at negative temperatures, in contrast to other processes which depend more clearly on the temperature."

l 351 Increase clarity "This fits the increase of riming, which was described for higher upward velocities as the saturation of the rising air can increase." → "This fits the increase of riming found for higher upward velocities, as the saturation of rising air can increase."

l 357 Increase clarity "...their tendency than others in respect to all thermodynamic variables evaluated here." → "their tendency than other processes in respect to all thermodynamic variables evaluated here."

Corrected subplot label references relating to Fig. 5 and Fig. 7 c and d switched in text. Added reference to C2 subplots.

Data availability statement:
Change of wording for clarity and added citation as the data set was now also mentioned in the text.

We have added a reference to the recently published paper by Omanovic et al. (2024, https://doi.org/10.5194/acp-24-6825-2024) as their results complement the discussion of our findings.